# 🖱 GROUNDING COMPUTER USE AGENTS ON HUMAN DEMONSTRATIONS

**Aarash Feizi**[1,2,4*]    **Shravan Nayak**[1,3*]    **Xiangru Jian**[5]    **Kevin Qinghong Lin**[6]    **Kaixin Li**[7]
**Rabiul Awal**[1,3,4]    **Xing Han Lù**[1,2]    **Johan Obando-Ceron**[1,3]    **Juan A. Rodriguez**[1,9]
**Nicolas Chapados**[4]    **David Vazquez**[4]    **Adriana Romero-Soriano** [1,2]    **Reihaneh Rabbany** [1,2]
**Perouz Taslakian** [1,2,4]    **Christopher Pal**[1,2,4,8]    **Spandana Gella**[4†]    **Sai Rajeswar**[1,3,4†]

[1]Mila - Quebec AI Institute    [2]McGill University    [3]Université de Montréal
[4]ServiceNow Research    [5]University of Waterloo    [6]University of Oxford
[7]National University of Singapore    [8]Polytechnique Montréal
[9]École de Technologie Supérieure    [10]CIFAR AI Chair

🌐 **Project Page:** https://groundcua.github.io

## ABSTRACT

Building reliable computer-use agents requires grounding: accurately connecting natural language instructions to the correct on-screen elements. While large datasets exist for web and mobile interactions, high-quality resources for desktop environments are limited. To address this gap, we introduce GROUNDCUA, a large-scale desktop grounding dataset built from expert human demonstrations. It covers 87 applications across 12 categories and includes 56K screenshots, with every on-screen element carefully annotated for a total of over 3.56M human-verified annotations. From these demonstrations, we generate diverse instructions that capture a wide range of real-world tasks, providing high-quality data for model training. Using GROUNDCUA, we develop the GROUNDNEXT family of models that map instructions to their target UI elements. At both 3B and 7B scales, GROUNDNEXT achieves state-of-the-art results across five benchmarks using supervised fine-tuning, while requiring less than one-tenth the training data of prior work. Reinforcement learning post-training further improves performance, and when evaluated in an agentic setting on the OSWorld benchmark using o3 as planner, GROUNDNEXT attains comparable or superior results to models trained with substantially more data. These results demonstrate the critical role of high-quality, expert-driven datasets in advancing general-purpose computer-use agents.

## 1 INTRODUCTION

The vision of computer-use agents (CUA) that operate software on behalf of users has gained significant momentum with recent progress in multimodal large language model–based agents (OpenAI, 2025; Anthropic, 2024; Qin et al., 2025; Wang et al., 2025a). These agents promise to automate routine work and make complex digital tools more accessible. For such agents to succeed, they must first *plan* the next step in a task, then *ground* the plan to the exact on-screen element to click, type, or drag. Accurate grounding is critical: without correctly identifying the right button or menu item, even a flawless plan cannot be executed. In FreeCAD, for instance, when asked to "open the color picker" (Figure 1), the agent must distinguish a small palette icon from look-alike tools, one of which it must precisely click. When grounding fails, the plan quickly veers off course, small errors compound, and tasks ultimately fail (Nayak et al., 2025). Moreover, grounding in desktop applications is challenging due to their complexity and diversity. These applications often feature high-resolution displays with dense layouts and visually similar elements, making precise localization difficult. Additionally, desktop applications can contain user-specific artifacts (e.g., documents or spreadsheets) that may not have been seen during training, adding variability and unseen contexts. Finally, collecting automated datasets for desktop environments with strong coverage is also challenging, as highlighted by recent datasets (Gou et al., 2024; Wu et al., 2024; Xie et al., 2025). To this

---

* Equal contribution.    † Equal supervision.

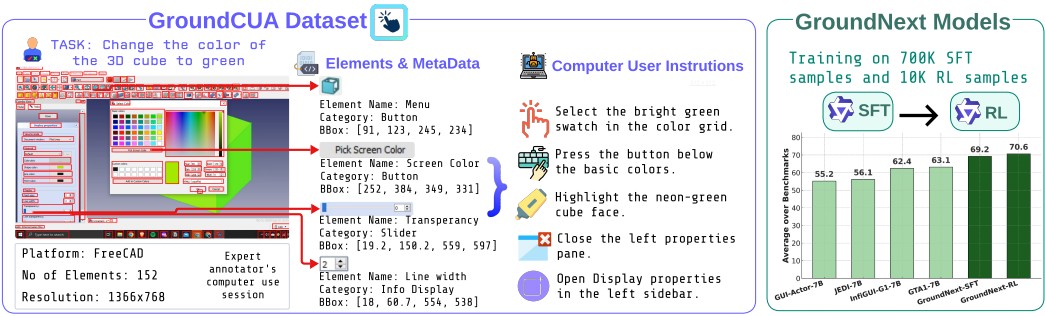

Figure 1: Overview of the GROUNDCUA dataset and GROUNDNEXT models. Human demonstrations of computer-use tasks are recorded as screenshots (example from FreeCAD) with UI metadata, which are processed into high-quality natural language instruction tasks for UI grounding. GROUNDNEXT is trained in two stages: SFT (700K samples) followed by RL (10K samples), achieving state-of-the-art grounding performance with efficient training.

end, we introduce **GROUNDCUA**, a large-scale, human-annotated dataset for desktop grounding. The dataset spans 87 applications across 12 categories, with 56K screenshots and 3.56M+ element annotations. These annotations are collected from task demonstrations by trained annotators, ensuring high-quality and densely labeled data that provides rich context for effective model training. It also reflects the pixel diversity of desktops, with resolutions ranging from 500K to 7M pixels and a substantial proportion of very small bounding boxes, highlighting the fine-grained challenges agents must overcome. Furthermore, GROUNDCUA includes fine-grained category information (menus, buttons, etc.) for 50% of the UI elements and includes multiple variants of related applications (e.g., LibreOffice and OnlyOffice), directly addressing the difficulty of similar yet distinct applications and enabling agents to learn robust, application-specific grounding strategies. Key highlights of GROUNDCUA compared to other datasets are: **Scale:** 56K annotated screenshots and 3.56 million elements; **Resolution, Element Size, and Density:** High-resolution images with maximum annotation density, covering almost every visible element, including small elements like icons and controls; **Expert Quality:** Human-verified annotations for high accuracy; **Application Diversity:** 87 desktop applications for broad real-world coverage. Using this dataset, we construct a 700K image-instruction pair instruction-tuning set that mimics real-world semantic interactions.

We introduce the **GROUNDNEXT** series of vision-language models, designed for precise grounding across desktop applications. The series includes models at 3B and 7B scales, offering a balance between efficiency and accuracy. Each model is trained in two stages: first, supervised fine-tuning (SFT) on 700K curated datapoints from GROUNDCUA, and second, reinforcement learning (RL) to further refine performance. This approach enables GROUNDNEXT to achieve state-of-the-art results on key desktop benchmarks, including ScreenSpotPro (Li et al., 2025), OSWorld-G (Xie et al., 2025), and UI-Vision (Nayak et al., 2025). Despite using significantly fewer SFT datapoints than state-of-the-art models like JEDI (which are trained on 9M datapoints), GROUNDNEXT outperforms existing models, demonstrating its **efficiency in training** and proving that high-quality, well-curated data can outperform larger, less precise datasets. In the RL stage, GROUNDNEXT further refines its grounding accuracy, achieving significant improvements **without relying on complex reward strategies**, unlike many RL-tuned models, which typically incorporate specialized reward functions and additional objectives. This shows the effectiveness of combining supervised fine-tuning (SFT) with high-quality data. This efficiency is further highlighted in agentic, multi-step tasks; evaluated on the OSWorld-Verified benchmark, GROUNDNEXT-3B not only significantly outperforms its 3B peers but also surpasses many larger models, including OpenCUA-72B and proprietary APIs. Notably, our 3B model achieves performance comparable to the much larger JEDI-7B, demonstrating significant practical utility for real-world, resource-constrained systems. Additionally, GROUNDNEXT excels in **cross-platform generalization**, delivering strong performance across desktop, mobile, and web environments; even though we only train on desktop dataset. Evaluated on benchmarks like MMBench-GUI (L2) and ScreenSpot-v2, in addition to desktop-specific tasks, GROUNDNEXT showcases its ability to generalize across a wide range of user interfaces and platforms. We plan to release both GROUNDCUAand the trained GROUNDNEXT models to support open research, providing a solid foundation for the development of reliable, adaptable computer-use agents across diverse environments.

In summary, our contributions are as follows:

- We introduce GROUNDCUA, a large-scale, human-annotated desktop grounding dataset with over 3.56 million annotations across 56K screenshots from 87 applications in 12 categories, providing dense, high-resolution, and fine-grained supervision for robust computer-use agents.

- We present the GROUNDNEXT series, vision-language models at 3B and 7B scales, trained on GROUNDCUA with SFT and RL, achieving state-of-the-art performance across desktop benchmarks and multi-step agentic tasks, with significantly fewer datapoints than prior models.

- We provide a comprehensive analysis of SFT and RL roles, evaluate our dataset's cross-domain impact and generalization beyond desktop, and study the benefits of open-source software for grounding performance. We release both GROUNDCUA and the GROUNDNEXT models to support open research. port open research.

## 2 RELATED WORK

**Computer-Use Agents.**  Recent advancements in computer-use agents have focused on enhancing their ability to understand and interact with user interfaces, ranging from simple commands to complex, multi-step tasks. Supervised fine-tuned models such as CogAgent (Hong et al., 2023), ShowUI (Lin et al., 2024), and Ferret-UI (You et al., 2024) have improved interaction capabilities by enabling zero-shot instruction-following across desktop, web, and mobile interfaces, combining vision, language, and action. Benchmarks like ScreenSpot-Pro (Li et al., 2025) and UI-Vision (Nayak et al., 2025) have emphasized the challenges of grounding natural language instructions in high-resolution desktop environments, particularly with dense screens and small elements. Grounding-focused agents, such as OS-ATLAS (Wu et al., 2024), UGround (Gou et al., 2024), and JEDI (Xie et al., 2025), have made significant progress by scaling training data to map language to specific UI elements. However, these methods often face challenges with data efficiency, particularly in complex desktop environments. Furthermore, recent RL-based approaches, inspired by DeepSeek-R1 (Guo et al., 2025), such as GUI-R1 (Luo et al., 2025), GUI-G$^2$ (Tang et al., 2025), and InfiGUI-G1 (Liu et al., 2025a), have addressed grounding through both simplistic and complex distance-based reward approaches. Despite these advancements, reliably grounding instructions to the correct on-screen elements remains a persistent bottleneck. To address this, we focus on high-quality, expert-annotated data to enhance grounding through both SFT and RL training, while prioritizing data-efficient fine-tuning to improve model performance.

Table 1: **Comparison of grounding datasets.** Columns: **H** = human-provided instructions and labels; **Desk** = includes desktop data; **E / Desk-E / S** = number of elements, desktop elements, and screenshots; **Res Range** = screenshot resolution range (MP); **EleArea** = average element area (% of screenshot); **#AvgE** = average elements per screen; **Perm** = permissive OSI-style license (e.g., Apache-2.0, MIT), **?** = not clearly reported. Datasets marked with * are grounding-specific versions constructed from the OS-ATLAS (Wu et al., 2024).

| Grounding Datasets | Annotation | | Scale | | | Avg. Data Stats | | | Perm? |
|---|---|---|---|---|---|---|---|---|---|
| | H | Desk | E | Desk-E | S | Res Range | EleArea | #AvgE | |
| UGround (Gou et al., 2024) | ✗ | ✗ | 9M | — | 773k | (0.4, 1.9) | — | 11.6 | ✓ |
| JEDI (Xie et al., 2025) | ✗ | ✓ | 4M | 2.4M | 575k | (0.9, 2.1) | — | 7.0 | ? |
| AGUVIS-G (Xu et al., 2024) | ✗ | ✗ | 3.8 M | — | 452k | (0.5, 2.1) | — | 8.5 | ? |
| OS-ATLAS (Wu et al., 2024) | ✗ | ✓ | 14.5M | 1.2M | 1.85M | (0.5, 5.2) | 0.53% | 7.8 | ✗ |
| RICOSCA*(Li et al., 2020a) | ✗ | ✗ | 170K | — | 18K | (0.5, 2.1) | 0.28% | 9.4 | ? |
| UIBert*(Bai et al., 2021) | ✗ | ✗ | 166K | — | 57K | (0.5, 2.1) | 0.24% | 2.9 | ✓ |
| Widget Caption*(Li et al., 2020b) | ✗ | ✗ | 101K | — | 14K | (0.5, 2.1) | 4.2% | 7.0 | ✓ |
| AMEX*(Chai et al., 2025) | ✗ | ✗ | 1.2M | — | 101K | (0.9, 4.5) | 2.1% | 11.8 | ? |
| SeeClick (Cheng et al., 2024) | ✗ | ✗ | 3M | — | 270K | (2.1, 2.1) | 0.33% | 11.2 | ? |
| AriaUI (Yang et al., 2024) | ✗ | ✓ | 4.1M | 150K | 295K | (1.3, 1.9) | — | 13.9 | ? |
| Fineweb*(Penedo et al., 2024) | ✗ | ✗ | 9.9M | — | 1.4M | (2.1, 2.1) | 0.29% | 6.9 | ✗ |
| **GROUNDCUA(ours)** | ✓ | ✓ | 3.56M | 3.56M | 55k | (0.4, 7.0) | 0.13% | 64.1 | ✓ |

**GUI Grounding Datasets.**  Training datasets for GUI grounding span mobile, web, and desktop platforms. Mobile datasets like RICO (Deka et al., 2017), UIBert (Bai et al., 2021), and

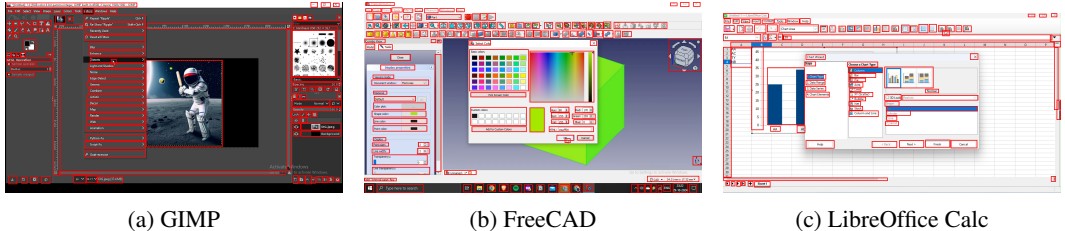

| (a) GIMP | (b) FreeCAD | (c) LibreOffice Calc |

Figure 2: Examples of screenshots from different applications in GROUNDCUA. Red bounding boxes indicate the annotated UI elements within each screenshot.

AMEX (Chai et al., 2025) provide element-level supervision within standardized layouts, simplifying extraction but limiting exposure to desktop-style density and iconography. Web-focused datasets, including SeeClick (Cheng et al., 2024) and UGround (Gou et al., 2024), scale grounding through automated harvesting from HTML/DOM, while Aguvis-G (Xu et al., 2024) broadens coverage across platforms. However, these automated pipelines likely overemphasize text-bearing elements while underrepresenting small icon-only controls, which are standard in desktop software. Desktop resources remain limited and challenging. OS-ATLAS (Wu et al., 2024) and AriaUI (Yang et al., 2024) assembles desktop splits via accessibility-tree traversal, yet accessibility signals are often incomplete or inconsistent, leading to missing or imprecise element labels (Muryn et al., 2025; Gou et al., 2024). JEDI (Xie et al., 2025) achieves scale through synthetic interface generation, but these simplified screens underrepresent genuine desktop complexity.

**How is ours different?** GROUNDCUA is the largest expert-annotated dataset for desktop grounding, comprising 55,568 screenshots across 87 open-source applications with over 3.56M human-verified UI elements. Compared to existing datasets, GROUNDCUA features denser screens, a wider resolution range, and smaller average element areas (see Table 1). It uniquely captures small desktop components, such as icons, toolbars, and controls, that are difficult to capture using automated tools. Its high-resolution images (ranging from 0.39 to 7.0M pixels) are substantially higher than other datasets and are the only ones to include very high-resolution images (see Figure 5, left). Additionally, GROUNDCUA features dense annotations that support semantics- and context-aware instructions, averaging 64 per screenshot, more than three times that of OS-Atlas (Desktop) and much higher than Aguvis-G (9) or UGround (11). Together, these properties make GROUNDCUA a comprehensive and challenging dataset for training robust desktop grounding agents.

## 3 GROUNDCUA DATASET

This section introduces GROUNDCUA, the largest and most diverse desktop-specific dataset annotated by human experts. We provide an overview of the data collection pipeline and annotation process, as well as our high-quality fine-tuning data below.

**Collecting demonstrations from human experts** We record real-world interactions of expert users performing tasks with desktop applications and annotated interface elements at scale. This approach captures user-driven interactions, resulting in a more realistic distribution of screenshots that better reflects real-world usage, compared to prior work that often relies on depth-first or breadth-first search to generate random interface states (Wu et al., 2024). Our pipeline consists of three main steps: selecting diverse applications, designing and executing practical tasks, and annotating screenshots. We partnered with a data labeling company for this process, with details on the annotator pool and training in Section A.2.

**Selecting diverse desktop applications** To support general-purpose computer-use agents, we selected 87 open-source applications across 12 categories (Table 5). Most applications are drawn from UI-Vision (Nayak et al., 2025), with four additional ones covering finance and scientific applications. By focusing on open-source applications with permissive licenses, we ensure the dataset can be freely released while encompassing a wide range of domains. These applications mirror the functionality of popular closed-source software (e.g., LibreOffice vs. Microsoft Office), making the dataset broadly applicable. Further details are provided in Section A.1.

**Designing and executing computer-use tasks.** We asked annotators to design everyday computer-use tasks that reflect common goals (e.g., drafting a document, editing a spreadsheet,

running a simulation) and then carry them out. This approach produces natural interaction trajectories, unlike random clicking, and yields screenshots that closely mirror real-world usage. In total, annotators completed over $10,000$ task demonstrations across $87$ applications [1].

**Dense annotation of screenshots.** From the recorded demonstrations, we extracted keyframes that capture the state of the interface immediately before a user action (e.g., a mouse click or text entry) that would trigger a change in the application. Annotators labeled every visible element in each keyframe using bounding boxes. For each element, they provided a textual label. This label was the element's name when available, the displayed text for shorter strings, or a concise summary in the case of long passages such as source code or detailed descriptions. We also extracted OCR using PaddleOCR (Cui et al., 2025) to extract raw text specifically for these longer segments. In addition, around 50% of the elements were assigned to one of eight high-level categories(see Table 6). In total, this process produced over 3.56 million annotated elements, making GROUNDCUA the largest and most diverse human-annotated grounding dataset for desktop environments to date. Examples of the annotations are provided in Appendix A.5, and further details of the annotation process are described in Appendix A.2.

**Constructing high-quality finetuning instructions** User queries in real-world settings can take various forms, from explicit references to UI elements (e.g., *Click 'Save'*), to functional commands (e.g., *Open a new tab*), or spatial descriptions (e.g., *Select the icon left of 'Files'*). To handle this diversity, we design a pipeline that leverages our dense annotations, which include bounding boxes, labels, categories, and OCR text, to construct diverse instruction-tuning data. These annotations enable the generation of highly contextual instructions, grounded directly in annotated screenshots. Unlike prior works that rely on pretrained models, our approach involves prompting a multimodal LLM with annotated bounding boxes, application names, element labels, and surrounding context. This ensures that the instructions are tightly linked to both the visual and textual content, making them semantically and contextually relevant. By leveraging nearly every visible element on the screen, we are able to create UI context-aware and challenging instructions. We generate three primary types of instructions: **Direct**, which describe an element's attributes, position, and surrounding context (e.g., *Click the magnifying-glass icon next to the search bar* for visual elements or *Click the button that has the text 'Save'* for OCR-based textual elements); **Functional**, which focus on the intended action of an element (e.g., *Open a new tab* instead of *Click the '+' button*); and **Spatial**, which guide the model based on the relative positioning of elements (e.g., *Click the element to the left of 'Files'* or *Select the icon between 'Undo' and 'Redo'*). We describe these instruction types in more detail in Section B and provide examples in Section B.6. These diverse instruction types, grounded in both visual and semantic context, provide a comprehensive foundation for training more effective and context-aware GUI agents.

**Dataset Statistics** GROUNDCUA consists of 56K screenshots, totaling 3.56 million annotated elements. On average, each screenshot contains 64 annotations, with some images having as many as $542$. The images have a mean resolution of 2.03 megapixels, with a range from $0.39$ to 7 megapixels. Bounding boxes are relatively small, covering just $0.13\%$ of the image area on average, underscoring the fine-grained nature of the annotations. This results in high-quality fine-tuning data, with 700K samples for SFT and 10K for RL, extracted from the densely annotated screenshots and metadata. Detailed distribution plots of resolution, bounding box sizes, and category-level statistics for both screenshots and annotations are provided in Appendix A.3.

## 4 TRAINING GROUNDNEXT MODELS ON GROUNDCUA

### 4.1 MODEL TRAINING

We use Qwen2.5-VL-Instruct as the base model for all experiments, considering both the 3B and 7B parameter variants. We finetune both the vision encoder and the language model, as preliminary experiments indicated that this leads to better grounding performance.

---

[1] We will release both the tasks and videos as part of the dataset.

**SFT** We first train the models with standard supervised finetuning. Training is performed on a single node with 8 H100 GPUs, using a global batch size of 128. Additional hyperparameter details are provided in Appendix C.2. For training data, we use the instruction tuning dataset introduced in Section 3. From this dataset, we use a subset of 700k instructions that balances coverage and diversity. This choice keeps the experiments practical and reproducible, while still being large enough to demonstrate the effectiveness of our dataset for grounding tasks. Further details on the composition of this subset, along with the choices made in its construction, are provided in Appendix C.1.

**RL Post-training.** In the next stage, we adopted RL post-training and explored several heuristics for constructing training data. GROUNDCUA allows us to sample from a much larger pool than the one used for SFT, so we selected 10K new elements not included in the original 700K SFT training set. This approach yielded the strongest generalization across benchmarks in our initial experiments, and we adopted it for the final model.

For policy optimization, we employed the Relative Leave-One-Out (RLOO) method (Ahmadian et al., 2024), which compares the reward of each rollout to the average reward of other samples within the same group, avoiding the need for training a separate critic model. Concretely, for a group of $n$ rollouts $\{y_1, \ldots, y_n\}$, the gradient is given by:

$$\nabla_\theta J(\pi_\theta) = \frac{1}{n} \sum_{i=1}^{n} \Big( R(y_i, x) - \frac{1}{n-1} \sum_{j \neq i} R(y_j, x) \Big) . \nabla_\theta \log \pi_\theta(y_i | x),$$

where $R(y_i, x)$ is the reward assigned to output $y_i$ given the input $x$. In our grounding setup, each $y_i$ corresponds to a sequence of tokens representing the predicted coordinates $(\hat{p}_i)$ on the image and $x$ corresponds to the input prompt and image.

**Reward Function.** We designed a customized discrete reward based on the normalized distance

$$R_{score}(\hat{p}, B, I) = \begin{cases} -1.0 & \text{if } \mathcal{D}_{norm} < -0.5, \\ -0.5 & \text{if } -0.5 \leq \mathcal{D}_{norm} < -0.1, \\ -0.1 & \text{if } -0.1 \leq \mathcal{D}_{norm} < 0, \\ 0.1 & \text{if } 0 \leq \mathcal{D}_{norm} < 0.1, \\ 0.5 & \text{if } 0.1 \leq \mathcal{D}_{norm} < 0.5, \\ 1.0 & \text{if } \mathcal{D}_{norm} \geq 0.5. \end{cases}$$

The normalized distance is defined as $\mathcal{D}_{norm} = \frac{\mathcal{D}(\hat{p}, B)}{\mathcal{D}_{ref}}$, where $\mathcal{D}_{ref} = \frac{\text{diam}(B)}{2}$ if $\hat{p} \in B$ and $\mathcal{D}_{ref} = \mathcal{D}_{max}(B, I)$ otherwise. $\mathcal{D}(\hat{p}, B)$ is the signed distance between the predicted coordinate $\hat{p}$ and the ground-truth bounding box $B$, with positive values inside. We use half the bounding box diameter if $\hat{p} \in B$ because that is the maximum distance a point inside $B$ can have from the boundary. This results in a $\mathcal{D}_{norm}$ value between -1 and 1.

This discrete scheme captures dominant error modes: predictions just outside the box receive a milder penalty, while predictions far outside receive a stronger one, and predictions inside the box are encouraged to move toward the center. We exclude reward model-based approaches due to the unreliable nature of current judges (Feizi et al., 2025; Lù et al., 2025). We experimented with alternative reward formulations (e.g., continuous and binary schemes), but ultimately adopted this discrete variant due to its superior empirical performance (see Appendix C.4 for details). We set the group size to $n = 8$, the batch size to 64, and trained for one epoch on a single H100 node (8 GPUs), consistent with the SFT setup.

## 4.2 EVALUATION

**Task Definition.** Given a screenshot $I$ and a user instruction $x$, the model predicts a 2D point $\hat{p} = (\hat{u}, \hat{v})$ in image coordinates. Let $B$ denote the axis-aligned ground-truth bounding box for the target element. A prediction is marked *correct* if $\hat{p} \in B$, and *incorrect* otherwise. We report the accuracy metric.

**Benchmarks.** We evaluate GROUNDNEXT on five key benchmarks that cover a wide range of grounding scenarios. For desktop applications, we use ScreenspotPro (Li et al., 2025), OSWorld-G

Table 2: **SFT-only results** on five challenging benchmarks. Results are shown for both 3B and 7B model scales. Only top-performing models are presented here; see Section D for full comparisons with additional baselines. Our GROUNDNEXT (SFT) consistently achieves the best average performance across all benchmarks, demonstrating the effectiveness of our high-quality data.

| Model | SSPro | OSW-G | MMB-GUI | SSv2 | UI-V | Avg |
|---|---|---|---|---|---|---|
| **≈ 3B** | | | | | | |
| Qwen2.5-VL-3B (Bai et al., 2025) | 16.1 | 27.3 | 60.8 | 80.9 | 6.3 | 38.3 |
| Qwen2.5-VL-3B (Agent mode) | 29.0 | 37.4 | 60.8 | 81.8 | 6.3 | 43.1 |
| PhiGround-4B-7C (Zhang et al., 2025) | 22.8 | 51.4 | 60.3 | 80.8 | 20.5 | 47.2 |
| JEDI-3B (Xie et al., 2025) | 36.1 | 50.9 | 66.5 | 88.6 | 18.7 | 52.2 |
| GUI-Actor-3B (Wu et al., 2025) | 42.2 | 48.9 | 69.8 | **91.0** | 19.7 | 54.3 |
| **GROUNDNEXT-3B (SFT)** | **48.6** | **62.2** | **75.5** | 87.3 | **58.2** | **66.4** |
| **≈ 7B** | | | | | | |
| Qwen2.5-VL-7B (Bai et al., 2025) | 26.8 | 31.4 | 33.9 | 88.8 | 0.9 | 36.4 |
| Qwen2.5-VL-7B (Agent mode) | 29.7 | 42.7 | 67.7 | 86.4 | 16.5 | 48.6 |
| OS-Atlas-7B (Wu et al., 2024) | 18.9 | 27.7 | 41.4 | 85.1 | 9.0 | 36.4 |
| UGround-V1-7B (Gou et al., 2024) | 16.5 | 36.4 | 65.7 | 87.6 | 12.9 | 43.8 |
| Aguvis-7B (Xu et al., 2024) | 39.5 | 38.7 | 45.7 | 86.0 | 13.7 | 44.7 |
| GUI-Actor-7B (Wu et al., 2025) | 44.6 | 47.0 | 70.9 | **92.1** | 21.9 | 55.3 |
| JEDI-7B (Xie et al., 2025) | 39.5 | 54.1 | 70.4 | 91.7 | 24.8 | 56.1 |
| **GROUNDNEXT-7B (SFT)** | **50.2** | **67.2** | **80.4** | 89.3 | **58.7** | **69.2** |

(Xie et al., 2025), and UI-Vision (Nayak et al., 2025), which focus on desktop interactions. To test cross-platform performance, we also use MMBench-GUI (L2) (Wang et al., 2025b) and Screenspot-v2 (Cheng et al., 2024), which include mobile and web splits in addition to desktop. This mix of benchmarks lets us evaluate performance across desktops, mobile, and web environments. Since UI-Vision overlaps with our dataset in platform coverage, we treat it as an in-domain benchmark, while the others are out-of-domain. We make efforts to minimize overlap during training, but due to annotation differences and the repetitive nature of desktop software, perfect separation isn't always possible.

**Baselines.** We compare GROUNDNEXT against two main types of baselines. First, we evaluate GROUNDNEXT (SFT) alongside several SFT-only variants to measure the impact of our instruction data (see Table 2). Then, we compare GROUNDNEXT (RL) with recent reinforcement learning-based models to assess the effectiveness of RL fine-tuning (see Table 3).

## 5 RESULTS

### 5.1 EFFICIENT SUPERVISED FINE-TUNING WITH HIGH-QUALITY DATA

We present the performance results of our models trained using SFT across five benchmarks in Table 2. Our models achieve the highest average performance for both 3B and 7B model sizes. For ≈3B, GROUNDNEXT-3B (SFT) ranks first on most datasets (except SSv2) and leads the SFT-only group by a clear margin with an average performance of **68.4** vs. 63.0 for the next best (GUI-Actor-3B) without considering UI-V, and **66.4** vs. 54.3 with UI-V (i.e., +5.4 and +12.1 points, respectively). Notably, our 3B SFT average also surpasses all *RL-tuned* 3B baselines. Adding the RL stage yields a small, consistent lift to **68.4 Avg / 70.0 (w/o UI-V)**, setting the best overall results in this size range. For ≈7B, GROUNDNEXT-7B (SFT) also leads among SFT-only models with **71.8 Avg** without UI-V and **69.2** with, outperforming the next best SFT baseline (JEDI-7B) by +7.9 and +13.1 points, respectively. Among RL-tuned systems, GROUNDNEXT-7B (RL) attains the top **Avg (w/o UI-V) = 73.0**. Overall, these results indicate the efficacy of our high quality data. Notably, our results are achieved with substantially less data and modest compute. We train on only 700K instructions, which is far below multi-million–sample corpora used by prior work (e.g., JEDI 9M). Yet, we outperform larger SFT baselines and remain competitive with RL-tuned systems. This suggests that

Table 3: **RL-tuned results**. We present results for the 3B and 7B model scales. We highlight the top-performing models here and refer readers to Section D for full comparisons with additional baselines. Our GROUNDNEXT(RL) achieves the highest average performance.

| Model | SSPro | OSW-G | MMB-GUI | SSv2 | UI-V | Avg |
|---|---|---|---|---|---|---|
| ≈ 3B | | | | | | |
| UI-R1-E-3B (Lu et al., 2025) | 17.8 | 48.8 | 68.4 | 88.6 | 16.5 | 48.0 |
| SE-GUI-3B (Yuan et al., 2025) | 35.9 | 46.1 | 66.3 | 86.8 | 15.0 | 50.0 |
| InfiGUI-R1-3B (Liu et al., 2025a) | 35.7 | 42.9 | 70.6 | 89.5 | 17.8 | 51.3 |
| GUI G$^2$-3B (Tang et al., 2025) | 36.4 | 53.5 | 66.3 | 87.6 | 18.7 | 52.5 |
| GUI-G1-3B (Zhou et al., 2025) | 37.1 | 49.5 | 71.0 | 89.5 | 20.3 | 53.5 |
| InfiGUI-G1-3B (Liu et al., 2025b) | 45.2 | 49.6 | 73.4 | **91.1** | 22.0 | 56.3 |
| GROUNDNEXT-3B (SFT) | 48.6 | 62.2 | 75.5 | 87.3 | 58.2 | 66.4 |
| **GROUNDNEXT-3B (RL)** | **49.8** | **64.2** | **77.1** | 88.8 | **62.1** | **68.4** |
| ≈ 7B | | | | | | |
| SE-GUI-7B (Yuan et al., 2025) | 47.3 | 33.9 | 34.5 | 68.9 | 16.7 | 40.3 |
| UI-TARS-1.5-7B (Qin et al., 2025) | 49.6 | 64.2 | 64.3 | 90.3 | 20.8 | 57.8 |
| GUI G$^2$-7B (Tang et al., 2025) | 47.5 | 61.9 | 79.5 | **93.3** | 25.6 | 61.7 |
| InfiGUI-G1-7B (Liu et al., 2025b) | 51.9 | 59.9 | 80.8 | 93.5 | 26.1 | 62.4 |
| GTA1-7B (Yang et al., 2025) | 50.1 | 67.7 | 79.4 | 92.4 | 25.7 | 63.1 |
| GROUNDNEXT-7B (SFT) | 50.2 | 67.2 | 80.4 | 89.3 | 58.7 | 69.2 |
| **GROUNDNEXT-7B (RL)** | **52.9** | **67.7** | **81.1** | 90.4 | **60.3** | **70.5** |

high-quality, densely grounded supervision and targeted instruction design can substitute for raw scale, delivering strong gains without escalating data volume or compute.

**GROUNDCUA compared to other SFT training corpora**    To make a fair comparison and highlight the quality of our dataset, we train the same base model (Qwen2.5-VL-3B-Instruct) on 100K samples from each of the following datasets: Aguvis, UGround, OS-Atlas (Desktop), JEDI, and GROUNDCUA. We use identical hyperparameters and preprocessing for all experiments (details in the Appendix). Figure 3 (yellow bars) summarizes the average performance across benchmarks, excluding UI-Vision. We observe that GROUNDCUA yields significantly higher SFT averages than all other training sources, demonstrating the benefits of its high-quality, densely grounded supervision.

## 5.2 REINFORCEMENT LEARNING POST-TRAINING RESULTS

RL post-training on top of the SFT models results in consistent but modest improvements for both the 3B and 7B models shown in Table 3. For the 3B model, **GROUNDNEXT-3B (RL)** achieves an average of **70.0** (without UI-V) and **68.4** overall, surpassing the SFT-only model, **GROUNDNEXT-3B (SFT)**, which scores **68.4** and **66.4**, respectively. For the 7B model, **GROUNDNEXT-7B (RL)** achieves **70.5**, improving upon **GROUNDNEXT-7B (SFT)**'s **69.2** (with UI-V). These results suggest that SFT, when trained with high-quality data, captures the majority of the model's performance, with RL offering targeted fine-tuning that provides incremental improvements. In practice, high-quality SFT can establish strong baselines, and RL can serve as an optional refinement step to further enhance performance. While our reward design is simple, we acknowledge that more sophisticated reward functions, such as those in Liu et al. (2025b), could lead to more substantial RL gains, which we leave for future work.

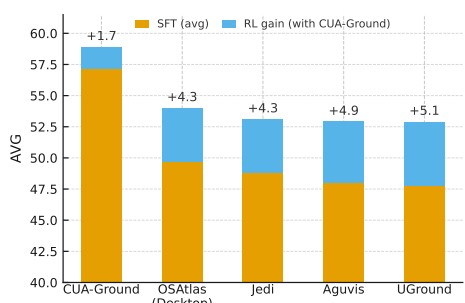

Figure 3: Mean SFT scores (orange) across benchmarks, with RL gains from 10k GROUNDCUA samples shown in blue.

Table 4: **Agentic performance comparison on OSWorld-Verified. Bold** and underline indicate the best-performing open-source model in each category. Our 3B model, GROUNDNEXT-3B, is among the top-performing open-source models, surpassing larger and proprietary models, highlighting its practical utility and efficiency for real-world agentic tasks.

| Model | OS | Office | Daily | Pro | Workflow | Overall |
|---|---|---|---|---|---|---|
| **Proprietary Models** | | | | | | |
| OpenAI o3 (OpenAI, 2025) | 62.5 | 14.5 | 21.4 | 38.8 | 16.5 | 23.0 |
| CUA (OpenAI, 2025) | 23.9 | 34.6 | 55.1 | 18.3 | 18.3 | 31.4 |
| Claude-4-Sonnet (Anthropic, 2025a) | 45.8 | 39.3 | 48.1 | 59.2 | 27.9 | 41.4 |
| Qwen3-VL-Flash (Bai et al., 2025) | 40.9 | 53.6 | 55.1 | 22.0 | 22.0 | 41.6 |
| UI-TARS-250705 (Qin et al., 2025) | 41.7 | 50.4 | 55.7 | 51.0 | 14.7 | 41.8 |
| Claude-4.5-Sonnet (Anthropic, 2025b) | 70.8 | 72.6 | 61.4 | 63.3 | 49.0 | 62.9 |
| **Open-source Models** | | | | | | |
| Qwen2.5-VL-32B (Bai et al., 2025) | 8.3 | 1.7 | 6.4 | 6.1 | 2.2 | 3.9 |
| Qwen2.5-VL-72B (Bai et al., 2025) | 16.7 | 4.3 | 6.4 | 2.0 | 3.2 | 5.0 |
| Kimi-VL-A3B (Kimi Team, 2025) | 12.5 | 6.0 | 21.7 | 18.4 | 1.1 | 10.3 |
| OpenCUA-A3B (Wang et al., 2025a) | 12.5 | 16.3 | 21.7 | 46.9 | 2.2 | 17.7 |
| UI-TARS-72B-DPO (Qin et al., 2025) | 37.5 | 19.0 | 34.6 | 63.3 | 8.3 | 27.1 |
| OpenCUA-7B (Wang et al., 2025a) | 41.7 | 22.2 | 37.1 | 49.0 | 9.3 | 27.0 |
| UI-TARS-1.5-7B (Qin et al., 2025) | 33.3 | 29.9 | 37.9 | 53.1 | 9.1 | 29.6 |
| OpenCUA-72B (Wang et al., 2025a) | 58.3 | **47.0** | 53.8 | 73.5 | 20.4 | 46.1 |
| JEDI-7B w/ o3 (Xie et al., 2025) | 50.0 | 46.1 | **61.9** | **75.5** | 35.3 | **51.0** |
| GROUNDNEXT-3B (RL) w/ o3 (ours) | **62.5** | **47.0** | 55.0 | 73.5 | **36.5** | 50.6 |

**Analyzing RL Gains.** We investigate the modest gains from RL (see Figure 3). In this setup, we start with SFT models (Qwen2.5-VL-3B-Instruct) trained on Aguvis, UGround, OS-Atlas (Desktop), JEDI, and GROUNDCUA, and then apply RL using 10K examples exclusively from GROUND-CUA. We find that models trained with GROUNDCUA during SFT show the smallest performance gains from RL, while models trained on other datasets benefit more from RL fine-tuning with GROUNDCUA. This suggests that SFT with GROUNDCUA already provides highly informative supervision, leaving fewer errors for RL to correct. Moreover, the magnitude of RL improvements correlates with the initial SFT performance: stronger SFT models yield smaller absolute gains because they start with fewer remaining errors. We explore this phenomenon in greater detail in Section D.6.

## 5.3 FURTHER ANALYSIS

**Agentic Performance** We evaluate GROUNDNEXT's performance in an agentic setting to assess its ability to ground in realistic, multi-step tasks. Experiments are conducted on the OSWorld-Verified benchmark using OpenAI o3 as the planner, which consumes task instructions and action history to generate grounding commands that GROUNDNEXT executes to locate target UI elements on the screen. Following the setup of (Xie et al., 2025; Yang et al., 2025; Wang et al., 2025a), we evaluate 361 tasks (excluding Google Drive–related ones) on an Ubuntu system with a 1920×1080 resolution, running on Microsoft Azure within 10 Docker environments.

The results in Table 4 highlight GROUNDNEXT-3B's strong performance. Within its 3B parameter class, GROUNDNEXT-3B (50.6 Overall) significantly outperforms peers like OpenCUA-A3B (17.7) and Kimi-VL-A3B (10.3). Notably, it surpasses many larger models, including OpenCUA-72B (46.1) and proprietary APIs such as Qwen3-VL-Flash (41.6) and Claude-4-Sonnet (41.4). The comparison with JEDI-7B, which also uses the o3 planner, is particularly notable. Despite being less than half the size, our 3B model achieves a comparable overall score (50.6 vs. 51.0) and demonstrates superior performance in 3 out of 5 categories (OS, Office, and Workflow). This performance from a compact 3B model underscores GROUNDNEXT-3B's significant practical utility, presenting it as an effective and efficient solution for real-world agentic systems where inference speed and resource constraints are critical factors.

**Gains from GROUNDCUA.** We investigate where GROUNDCUA yields the greatest gains by studying the performance of GROUNDNEXT. Since GROUNDCUA primarily covers desktop software, we expect the largest gains on desktop benchmarks. Our results confirm this: GROUND-NEXT-7B (RL) achieves the best performance on UI-V, OSW-G, and SSPro. For mixed datasets such as MMBench-GUI, GROUNDNEXT shows a 3.66% improvement on desktop platforms over the second-best model, InfiGUI-G1, with notable gains coming from Linux and macOS (see Section D.3). At the element level, the most significant improvements are observed in icon recognition. For example, on SSPro, we outperform most models by an average of 10.7% in icon recognition (see Table 11). This reflects the high density of icons in desktop applications and suggests that the diversity in GROUNDCUA provides richer knowledge, leading to better performance on icons.

**GROUNDNEXT generalization across domains.** Next, we evaluate the generalization ability of GROUNDNEXT, trained primarily on desktop software, to mobile and web interfaces using SSv2 and MMBench-GUI. On MMBench-GUI, GROUNDNEXT-7B (RL) performs competitively across both domains, achieving 89.2% on mobile and 81.9% on web, compared to the next best model, i.e., InfiGUI-G1-7B, at 90.9% and 85.3%, respectively. On SSv2, GROUNDNEXT achieves comparable results on mobile but falls behind on web. A detailed error analysis is provided in Section E. These results suggest that while GROUNDCUA enables strong cross-domain generalization, future work could explore augmenting desktop data with web and mobile sources to further enhance performance.

**Effects of using open source applications.** To study the impact of open-source software, we examine SSPro performance across various categories, focusing particularly on icon recognition. Icons often require application-specific knowledge, unlike text, which is more general in nature. As shown in Table 11, GROUNDNEXT achieves the best icon performance in the *Office Suite*, *Development*, *Creative*, *Scientific*, and *CAD* categories, and ranks second in *OS*. The presence of open-source office software, such as LibreOffice, likely contributes to the strong results in the *Office Suite* category. Similarly, the diversity of open-source development tools and creative software, such as video and image editing programs, results in significant improvements, with our model outperforming the next best model, i.e., InfiGUI-G1-7B, by 15.9% in *Development* and 8.4% in *Creative* for icon accuracy. Future work could further analyze the impact of application similarity to determine whether applications more similar to those in our dataset lead to higher performance.

## 6 CONCLUSION & DISCUSSION

We introduced GROUNDCUA, a human-annotated desktop grounding dataset spanning 87 applications (56K screenshots, 3.56M+ elements) with dense keyframe labels that reflect real interaction states. From these annotations, we constructed real-world computer-use instruction tasks for grounding. We developed the GROUNDNEXT family of models and following recent trends, trained it first with SFT and then RL on verifiable rewards. Across five challenging benchmarks, GROUNDNEXT achieves state-of-the-art results despite using substantially less SFT training data than many prior works. The key takeaway is that high-quality data drives reliable desktop grounding more effectively than sheer data volume. By releasing both the dataset and other research artifacts, we aim to unlock grounding as a core capability, laying the foundation for end-to-end computer-use agents that can perform complex tasks across diverse desktop applications.

While this work advances desktop grounding and demonstrates the value of high-quality expert demonstrations, it also opens up new opportunities and raises important questions. First, we train models with limited scale and compute, but the dataset can support variable-sized fine-tuning sets to further scale model performance. Second, our dense annotations should enable the development of precise and expressive reward signals for RL, moving beyond the simplistic one used in this paper. This creates opportunities to systematically study how different reward designs impact grounding accuracy. Third, cross-domain generalization remains a key frontier. Desktop environments involve complex, multi-window workflows, whereas mobile and web tasks are lighter and more context-specific. Mixing data across these domains could yield models that operate seamlessly across platforms, though balancing these domains and addressing transfer bottlenecks will require careful study. Finally, GROUNDCUA includes platform- and category-level metadata, enabling research on continual learning and adaptation, evaluating how agents adapt to unseen applications and continually improve as new interaction paradigms emerge.

## ETHICS STATEMENT

Our work focuses on the responsible development of computer-use agents through transparent dataset curation and model training. We have taken significant steps to protect user privacy by ensuring all desktop applications used are open-source with permissive licenses, and no personally identifiable information (PII) was collected during screenshot annotation. All human annotators were fairly compensated and worked under proper data protection protocols. While we have filtered potentially harmful content from our dataset, we cannot fully guarantee that models trained on GROUNDCUA will not generate inappropriate instructions or interact with sensitive interface elements inappropriately. Users and developers are strongly encouraged to implement appropriate safeguards and human oversight when deploying computer-use agents in production environments. Additionally, all human evaluation studies were conducted by collaborating researchers following established ethical guidelines, with no PII collected during the evaluation process.

We disclose that there are no conflicts of interest that would bias this work. Any funding sources will be listed in the camera-ready version per ICLR policy.

## REPRODUCIBILITY STATEMENT

We are committed to ensuring full reproducibility of our work by providing comprehensive implementation details and releasing all necessary resources. All artifacts, including the complete GROUNDCUA dataset, GROUNDNEXT model weights, training code, evaluation scripts, and detailed data sheets, will be publicly released. We have thoroughly documented all hyperparameters, training procedures, data preprocessing steps, and evaluation metrics in the appendices to enable accurate replication of our results. The human annotation guidelines, quality assurance protocols are fully described to ensure transparency in dataset creation. The instruction generation prompts, model architectures, and benchmark evaluation procedures are detailed to facilitate consistent reproduction across different research groups.

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

# APPENDIX

## Table of Contents

## A   GROUNDCUA – CREATION

### A.1   PLATFORMS

Table 5: Categories of desktop applications and their corresponding applications.

| Category | Platforms |
|---|---|
| Education | Anki, Zotero, Calibre, OpenBoard, Mendeley |
| Browsers | Brave, Chromium, Mozilla Firefox, DuckDuckGo |
| Development | VSCode, Atom, Eclipse, NetBeans, PyCharm, IntelliJ IDEA, Brackets, Geany, Bluefish, KDevelop, Komodo Edit, Code::Blocks, Qt Creator, Arduino IDE |
| Productivity | LibreOffice Calc, LibreOffice Draw, LibreOffice Impress, LibreOffice Writer, draw.io, Joplin, OpenProject, Affine, PDFedit, OnlyOffice Calendar, OnlyOffice Document Editor, OnlyOffice Forms, OnlyOffice PDF Forms, OnlyOffice Presentation, OnlyOffice Spreadsheet, Nextcloud, Gnumeric, Simplenote, WeKan |
| Graphics and Design | Blender, GIMP, Inkscape, Krita, darktable, FontForge, Scribus, WordPress |
| Video and Audio Production | OpenShot, OBS Studio, Lightworks, Shotcut, Natron, OpenToonz, Audacity, MuseScore |
| Communication | Element, Signal, Mastodon, Lemmy, Matrix, Zulip, Jitsi |
| Entertainment | VLC Media Player, Kodi, Emby |
| System Utilities | Ubuntu Terminal, Conky, Bash, 7-Zip, Flameshot, Nemo, gedit |
| Security | Bitwarden, Cryptomator |
| Finance and Business Analytics | GnuCash, Frappe Books, Metabase |
| Scientific | RStudio, Veusz, GNU Octave, GrassGIS, QGIS, FreeCAD, Spyder |

We select 87 platforms, focusing on open-source software with permissive licenses. These applications span 12 diverse categories, detailed in Table 5. Our selection is motivated by the under-representation of such platforms in existing datasets and the flexibility provided by permissive licensing, which enables dataset release with minimal restrictions. We primarily rely on UI-Vision (Nayak et al., 2025) as the source for platforms, as they motivated their platform selection similarly. We additionally include 4 platforms to improve coverage across finance and scientific categories. We further show that this choice does not compromise generalization (see Section 5.3), as the open-source software usually shares UI elements and layout with its closed-source counterparts. For instance, LibreOffice and Office Suite share many interface elements, layout, and functionality. This ensures broader applicability of GROUNDCUA.

### A.2   HUMAN ANNOTATION

We collaborated with a professional data labeling vendor that specializes in dataset curation for AI applications. The annotation effort spanned three phases, beginning with a pilot study where we worked closely with the annotation team to refine task instructions and provide iterative feedback. The annotation team consisted of around 70 individuals, organized into multiple tiers of annotators, quality assurance specialists, and project managers. The majority of the team was located in India and Latin America, with participants in the 20–35 year age group and a balanced gender distribution. All annotators held at least a bachelor's degree in technical fields such as Computer Science or Engineering and had prior experience in data labeling and user interface research.

Table 6: UI element categories in GROUNDCUA with descriptions and representative examples.

| Category | Description and Common UI Elements |
|---|---|
| **Input Element** | Interactive fields where users enter or modify data, like text boxes, checkboxes, radio buttons, etc. |
| **Sidebar** | Vertical or horizontal panels that provide quick access to tools or navigation. Examples include tool palettes, folder trees, settings sidebars. |
| **Information Display** | Regions that primarily present textual or numerical information. Examples include labels, console outputs, document text, and code blocks. |
| **Button** | Clickable controls that trigger an action like submit button, "OK/Cancel" buttons, play/pause buttons |
| **Navigation** | Elements that help users move within or across applications. Examples: tabs, back/forward arrows etc. |
| **Visual Elements** | Non-textual graphical elements that convey information or functionality. Examples include icons, thumbnails, images, charts, and progress bars. |
| **Menu** | Structured lists of commands or options, often hierarchical. Examples: file menu, context menu, dropdown menus. |
| **Others** | Elements not covered by the above categories, often decorative or container elements like spacers. |

Annotators underwent a training process to become familiar with the platforms and annotation guidelines. They were compensated hourly, with each task requiring on average 60–90 minutes to complete, including quality checks. The process began with the creation of computer-use tasks for 87 software applications (see Table 5). Annotators then executed these tasks while screen recordings were collected. From these recordings, we extracted keyframes corresponding to major user interactions. Each keyframe was annotated using a custom tool, where annotators drew bounding boxes around all visible interface elements. For each bounding box, annotators assigned a label corresponding to the element's name, or, in the case of textual elements, the text was also provided in addition to the element name. For long text segments such as source code or lengthy descriptions, annotators provided a concise summary that captured the main theme. To supplement these summaries, we also applied OCR using PaddleOCR (Cui et al., 2025) to extract the full text when available. In addition, every element was assigned to one of six high-level categories. We applied rigorous quality assurance at multiple stages. Annotations were reviewed by dedicated quality specialists, cross-checked by the authors, and validated using custom evaluation scripts. This pipeline allowed us to construct a large-scale dataset of grounded user interface interactions with high diversity and reliable annotation quality.

### A.3 DATASET STATISTICS

We provide detailed statistics for GROUNDCUA. Figure 4 presents the overall dataset statistics. Figure 4a shows the number of annotations across the 12 categories, while Figure 4b reports the number of screenshots per category. We also analyze the pixel distribution of screenshots in Figure 4c, observing a wide range from roughly 0.3 megapixels to 7 megapixels. The distribution of bounding box areas, shown in Figure 4d, highlights the prevalence of small UI elements in the dataset. Finally, Figure 4e shows the number of bounding boxes per screenshot, with some screenshots containing up to 500 annotated elements and Figure 4f shows the distribution of desktop applications across 12 different categories.

### A.4 COMPARISION WITH PRIOR WORKS

**Comparative Analysis with Existing Datasets** We compare GROUNDCUA against four recent grounding datasets: UGround (Gou et al., 2024), Aguvis (Xu et al., 2024), OS-Atlas (Wu et al., 2024), and JEDI (Xie et al., 2025). For OS-Atlas and JEDI, which are much larger, we sample 200k images for screenshot-level analysis, with bounding-box statistics computed over all annotations. As shown in Figure 5 (left), GROUNDCUA's screenshots range from 0.5M–7M pixels, averaging 2.0M, capturing high-resolution desktop environments. UGround and OS-Atlas (Desktop) have lower resolutions (1.1M and 1.6M), limiting their detail. Figure 5 (right) highlights GROUNDCUA's smaller median element size, with many fine-grained targets like icons and small controls, typical of desktop

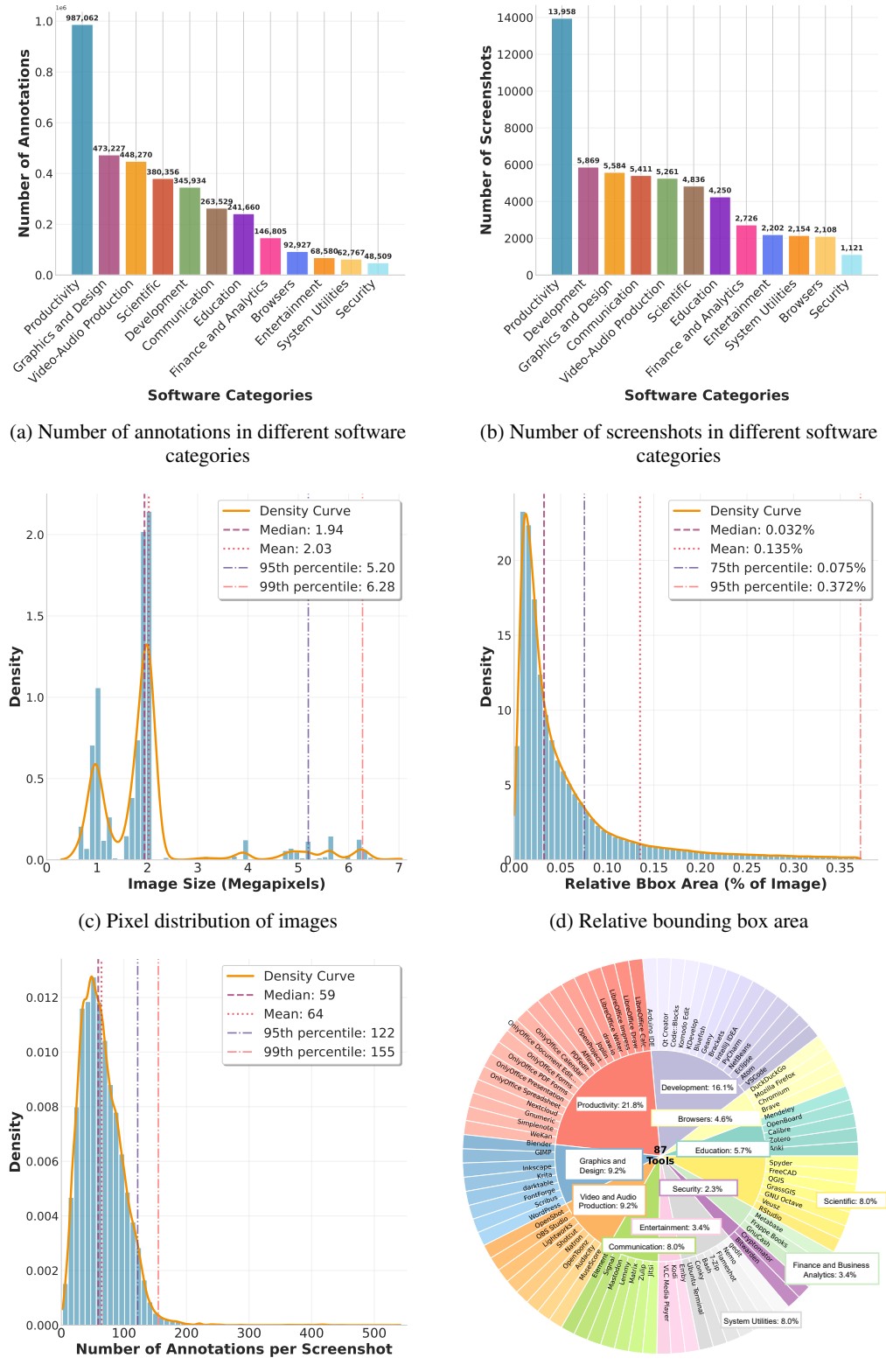

(a) Number of annotations in different software categories

(b) Number of screenshots in different software categories

(c) Pixel distribution of images

(d) Relative bounding box area

(e) Distribution of number of annotations in an image

(f) Desktop application across different categories

Figure 4: Dataset Statistics

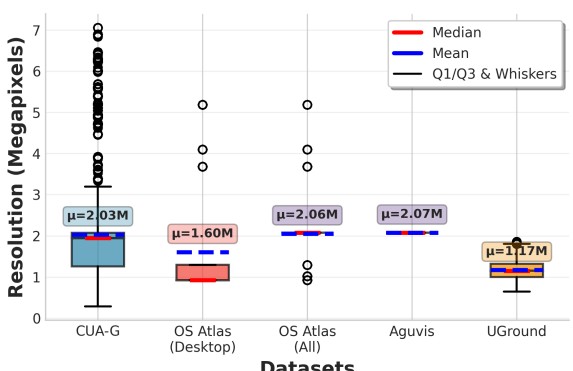 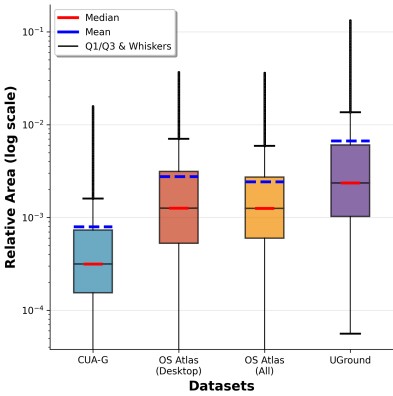

Figure 5: Comparison across different datasets. (Left) Pixel distribution for different datasets. (Right) Relative bounding box area in log scale.

interfaces. In contrast, other datasets focus on larger, more salient elements. GROUNDCUA also has denser annotations, averaging 64 per screenshot, more than three times that of OS-Atlas (Desktop) and much higher than Aguvis (9) or UGround (11). JEDI, despite its scale, has sparser annotations due to its reliance on synthetic data. UGround and Aguvis cover web interfaces, while OS-Atlas uses automated accessibility-tree traversal, which is often incomplete and prone to errors (Gou et al., 2024; Muryn et al., 2025), resulting in less precise annotations. JEDI is impressive in scale but lacks dense, real-world coverage due to the synthetic pipeline involved in creating the dataset. GROUND-CUA, with its high-resolution, human-verified annotations, and extensive platform diversity, fills a crucial gap by providing a more accurate and detailed representation of desktop environments.

## A.5 DATASET EXAMPLES

Figure 6 shows examples of screenshots from several software platforms with bounding boxes overlaid on the images.

## B INSTRUCTION TUNING DATA

GROUNDCUA contains over 3.5M annotated elements. Desktop screens are highly redundant, with many UI elements repeating across views. To reduce duplication before building instructions, we deduplicate elements using text matching on labels and perceptual image similarity (pHash) computed on crops defined by each element's bounding box. This produces roughly 900k unique elements. We use strict thresholds during filtering. While this may remove some valid cases, it yields a diverse, non-redundant pool overall. We also randomize selection across screenshots so that no single interface is over-represented. The filtered elements form the base for constructing the instruction tuning data. We detail the different types of instructions we have created below and provide examples in Figure 7.

### B.1 DIRECT INSTRUCTIONS

Direct instructions explicitly refer to the element (Click the "File" button) that the model should act on. These are the most common types of instruction a CUA would encounter. We first create a class of descriptive instructions for every element, which incorporates attributes such as color, shape, position, and nearby context. These descriptions provide richer context for the model and help reduce ambiguity. We generate these instructions by prompting Qwen2.5-VL-72B with the element's bounding box, platform name, annotated label, the full screenshot, and an optional zoomed crop. We also ask the model to provide the location of the element if there are other similar elements to disambiguate. We additionally use category information to create three specific types of direct instructions:

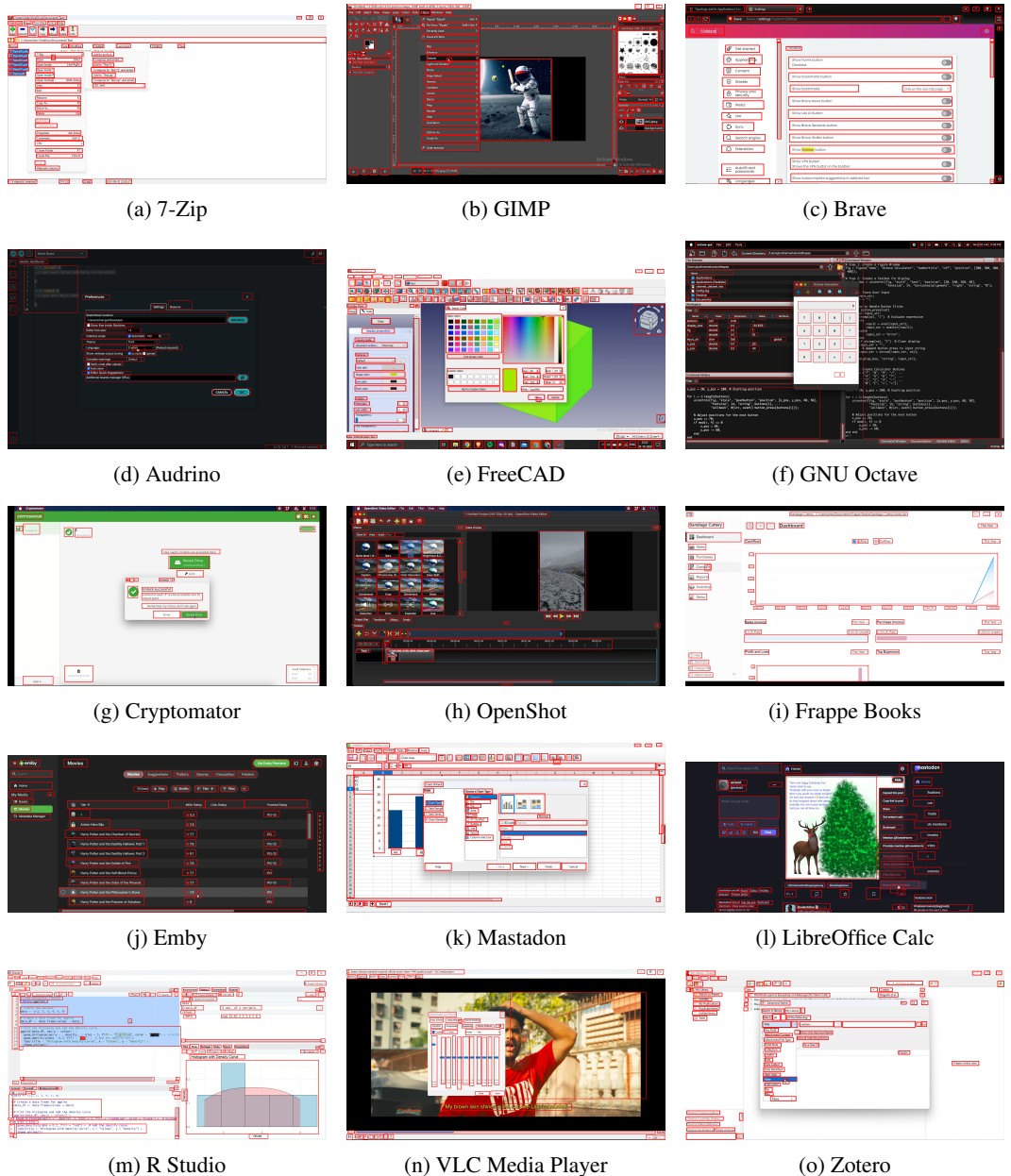

(a) 7-Zip      (b) GIMP      (c) Brave

(d) Audrino      (e) FreeCAD      (f) GNU Octave

(g) Cryptomator      (h) OpenShot      (i) Frappe Books

(j) Emby      (k) Mastadon      (l) LibreOffice Calc

(m) R Studio      (n) VLC Media Player      (o) Zotero

Figure 6: Examples of screenshots from different platforms in GROUNDCUA. Red bounding boxes indicate the annotated UI elements within each screenshot.

**Description Instruction Prompt**

You are an expert UI analyst. You are given a screenshot with a target element in a red bounding box, a cropped image containing the target element in a red bounding box, the name of the element and the platform name.

Can you find it? Is it visible from the screenshot? Can you write a concise description that is sufficient for humans to locate it from the screenshot? The response must be relevant to the platform and element name provided. Do not reference the red bounding box and that it is highlighted.

If you find other identical elements, your description must include specific details about the

target element's location and other unique attributes to differentiate it from the others.
Only output what you are sure about. Do not make assumptions. Return the response in the
following JSON format:
{
"visible": true,
"description": "your description here"
}

Platform: {platform name}
Target Element Label: {text}

**Textual elements.** We identify textual elements by matching OCR output with the human-annotated label and by selecting items from the *Information Display* category. We then embed the extracted text into about 100 templates that directly instruct the model to move to these labels. Some templates used to generate instructions are provided below.

**Textual Elements Instruction Templates**

1. Do you see the text 'text'? Please click on it.
2. Please locate the user interface component marked with the text 'text' and then proceed to click on it.
3. Make your way to the 'text' label with your cursor.
4. You are required to find the element associated with the text 'text' and then move your cursor to hover over it.

**Visual elements.** For icon-based or other visual elements (e.g., tool icons, shapes, images), we generate concise captions that highlight distinctive features and local context (e.g., "Click the magnifying-glass icon next to the search bar"). These are produced using Qwen2.5-VL-72B by providing the element crop, its bounding box in the full screenshot, the platform name, and the annotated label.

**General templates.** In addition to text and visual elements, we design a set of general instructions that apply to any element. These are created heuristically using about 120 templates (e.g., "Click on the following element:") or generated by prompting an MLLM.

**General Instruction Prompt**

You are an expert UI analyst. You are given a screenshot with a target element in a red bounding box, a cropped image containing the target element in a red bounding box, the name of the element and the platform name.
Is it visible from the screenshot? Generate a concise, imperative instruction a user would give to operate or interact with the target element.
The response must be relevant to the platform and element name provided. Do not reference the red bounding box and that it is highlighted.
If you find other identical elements, your description must include specific details about the target element's location and other unique attributes to differentiate it from the others.
Only output what you are sure about. Do not make assumptions. Return the response in the following JSON format:
{
"visible": true,
"instruction": "your description here"
}

Platform: {platform}
Target Element Label: {text}

## B.2 FUNCTIONAL INSTRUCTION PROMPT

Functional instructions describe an element by its purpose rather than its name (e.g., "Open a new tab"). We focus on *Buttons* and *Menus* since these most often encode actions. For each candidate element, we prompt Qwen2.5-VL-72B with the full screenshot, the element crop and bounding box, platform name, and the annotated label, asking for a concise functional instruction (e.g., "Open a new tab").

---

**Functional Instruction Prompt**

You are an expert UI analyst. You are given a screenshot with a target element in a red bounding box, a cropped image containing the target element in a red bounding box, the name of the element and the platform name. Is it visible from the screenshot? Generate a task-oriented instruction that describes a user's goal. The instruction must implicitly identify the target element by describing what it helps the user accomplish (not the name of the element).
The response must be relevant to the platform and element name provided. It should also be concise and to the point. Do not reference the red bounding box and that it is highlighted. Include the location or other unique attributes if there are other identical elements.
Only output what you are sure about. Do not make assumptions. Return the response in the following JSON format:
{
"visible": true,
"function": "your description here"
}

Platform: {platform}
Target Element Label: {text}

---

## B.3 SPATIAL INSTRUCTIONS

Spatial instructions locate a target element by its position relative to another element (anchor), using relations such as *left*, *right*, *above*, *below*, and *between*. We leverage dense annotations to choose anchors that are close to the target and have reliable labels (e.g., "Click the icon to the left of 'Files'"). We generate these with simple templates that insert the anchor's label and the relation. Some templates used to produce instructions are provided below.

---

**Spatial Instructions Templates**

1. Place your mouse on the element directly to the right of "{element}".
2. Hover your mouse on the element immediately to the left of "{element}".
3. Hover your mouse on the element between "{element_1}" and "{element_2}".
4. Place your mouse on the element directly above "{element}".

---

## B.4 EXAMPLES

Figure 7 shows different kinds of instructions generated by our data generation pipeline.

## B.5 NEED AND IMPACT OF DIFFERENT INSTRUCTION TYPES

As mentioned above we have three main instruction types: Direct, Functional, and Spatial. The "Direct" category is itself broad, encompassing instructions based on descriptions, text/visuals, as well as general/heuristic templates. To analyze their impact of different instruction types and their subtypes, we sampled 100k datapoints for each distinct instruction subtype and trained Qwen2.5-VL-Instruct-3B model. The results are presented in Table 7.

As shown in Table 7, Functional instructions yield the highest average performance. We hypothesize this is because these goal-oriented instructions (e.g., "Open a new tab") closely match the tasks in

Table 7: Ablation study on different instruction types. We sample 100k data points for each type and train a Qwen2.5-VL-3B model.

| Data Type | SS-Pro | OSW-G | UI-V | MMB-GUI | SSv2 | Avg |
|---|---|---|---|---|---|---|
| Functional | 43.0 | 51.0 | 27.6 | 73.4 | 88.1 | **56.6** |
| Direct - Description | 36.5 | 58.5 | 24.5 | 70.1 | 85.0 | 54.9 |
| Direct - General templates | 37.9 | 52.8 | 26.4 | 73.8 | 86.1 | 55.4 |
| Direct - Text and visual | 35.3 | 51.0 | 20.3 | 64.2 | 85.1 | 51.2 |
| Direct - Miscellaneous | 32.5 | 56.3 | 24.7 | 67.1 | 85.4 | 53.2 |
| Spatial | 20.5 | 52.5 | 22.4 | 67.8 | 74.1 | 47.5 |

the evaluation benchmarks. However, the other instruction types are crucial for building a robust, well-rounded agent for two main reasons:

1. **Complementary Strengths:** While "Functional" is best on average, other types excel at specific tasks. For example, in the OSWorld-G text-recognition category, the "Direct - Text and visual" split achieves a 0.64 score, outperforming the "Functional" split's 0.60.

2. **Preventing Overfitting:** We observed that models trained on only one instruction type become brittle. For example, a model trained only on "Description" instructions sees its performance drop by 4% on ScreenSpot-Pro when the prompt "Click on the element with the following description:" is not prepended to the benchmark's instructions.

Our final 700K SFT dataset is a chosen mix to ensure the model is both high-performing and less sensitive to prompts used. Hence, when evaluating our final models, we do not provide any prefix prompts and evaluate directly on the instructions provided by the benchmarks.

### B.6 QUALITY OF INSTRUCTION TUNING DATA

Our MLLM pipeline for generating the instructions is highly robust for a key reason: we are not dependent on the MLLM's open-ended knowledge. Our pipeline is highly constrained. It provides the MLLM with strong, ground-truth context, including the element's name, its bounding box, and additional screenshot context (e.g., elements around the target element for spatial instructions). The task is one of grounded rephrasing or description, not open-ended task creation. This also clearly reflects the outcome of the trained model using generated instructions across a wide variety of benchmarks and the reported result in the paper.

To further verify this systematically, we performed a human evaluation on the generated instruction-tuning set. Three annotators who are not the authors of the paper annotated 100 randomly selected instructions to check for validity (i.e., whether the instruction accurately describes the element being grounded). Using a majority vote to aggregate these annotations, we found an error rate of 4%. We believe this represents a low error rate for a training dataset, highlighting the quality and reliability of our MLLM-generated instruction data.

## C TRAINING

In this section, we describe the training process for GROUNDNEXT. We outline the key design choices behind our SFT and RL setups, including data selection and filtering strategies, hyperparameter configurations, and other relevant details. We also report experimental observations, highlighting the impact of these choices and the insights gained during development.

### C.1 SFT DATA

From the instruction-tuning corpus, we curate a split of 700K size with 50% direct instructions, 35% functional instructions, and 15% spatial instructions.

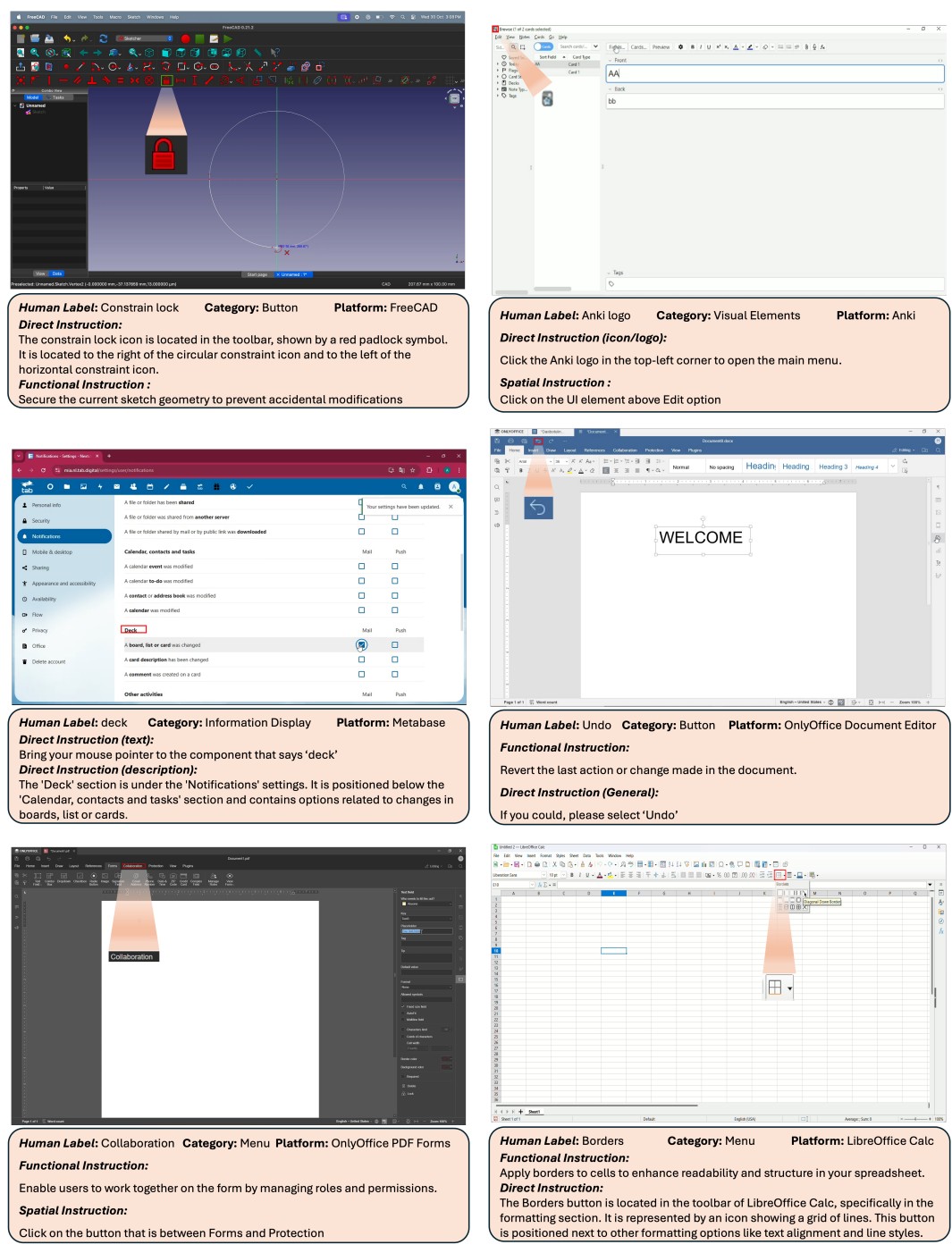

Figure 7: Instruction tuning data examples.

## C.2 SFT Training Details

We use LlamaFactory (Zheng et al., 2024) to train our SFT models with a learning rate of 3e-6, cosine decay, and a warmup ratio of 0.05. Models are trained for two epochs, as this consistently outperforms training for a single epoch. Preliminary experiments also show that training the entire model, rather than only the LLM, is more effective; we therefore adopt this configuration throughout. All models are trained on a single H100 node with 8 H100 GPUs, using a global batch size of 128, gradient accumulation of 16, and a per-device batch size of 1.

## C.3 RL DATA

For RL training, we first performed rejection sampling on the SFT training set using the SFT model itself. Specifically, we extracted the model's errors and sampled 10K instances, which were then used to run RL. While this yielded modest improvements, the SFT model was already strong, and many of the extracted errors corresponded to noisy or ambiguous datapoints (e.g., prompts with multiple valid answers or inconsistent labels). These issues limited the effectiveness of this approach.

We next applied RL on top of the SFT model using 10K previously unseen samples from GROUND-CUA. This strategy avoided noise from ambiguous training points and yielded a more significant performance boost. Consequently, our final setup exclusively used the 10K samples unseen during SFT from GROUNDCUA.

We also explored incorporating a small amount of out-of-distribution data to encourage generalization to web and mobile domains. Specifically, we added 10K samples from GUIAct (Chen et al., 2024), in addition to 10K samples from GROUNDCUA, split evenly between mobile (5K) and web (5K). Unlike the gains observed when adding in-distribution samples from GROUNDCUA, this preliminary attempt did not yield consistent improvements. We note, however, that our setup was limited in scope and did not include rejection sampling or other analysis. A more systematic investigation of combining our dataset with complementary sources, particularly in the context of RL training to improve cross-platform performance, is an exciting direction for future work.

## C.4 RL TRAINING DETAILS

For our RL training, we compared two rule-based optimization methods, Group Relative Policy Optimization (GRPO) and Relative Leave-One-Out (RLOO). Empirically, and as pointed out in previous literature (Zhang et al., 2025), we found that RLOO produced more stable learning and better results. The RLOO objective can be written as:

$$\nabla_\theta J(\pi_\theta) = \mathbb{E}_{\tau \sim \pi_\theta} \left[ \sum_{t=1}^{T} \nabla_\theta \log \pi_\theta(a_t|s_t) \left( R(\tau) - \frac{1}{n-1} \sum_{j \neq i} R(\tau_j) \right) \right], \quad (1)$$

where $R(\tau)$ is the reward of trajectory $\tau$, and the baseline is computed as the average reward of all other trajectories in the same group (excluding the $i$-th trajectory). This avoids training a critic model and instead uses relative group comparisons. In our case, the trajectories are the predicted coordinates by the model, and the reward is defined based on where the predicted point is relative to the bounding box. For the grounding task, $\tau$ is a sequence of tokens, which represents the predicted coordinate.

**Reward Formulation.**

1. *Continuous reward:* Based on the normalized distance $d$ between the predicted point $\hat{p}$ and the ground-truth bounding box $B$, we defined:

$$r = 1 - d, \quad d = \frac{\|\hat{p} - p^*\|}{\text{MaxDist}(B, W, H)},$$

where $p^*$ is the closest point in $B$, and $\text{MaxDist}(B, W, H)$ is the maximum possible distance a point inside an image of width $W$ and height $H$ can have. However, this suffered from sparsity and weak gradient signals.

2. *Binary reward:* A simple scheme assigning

$$r = \begin{cases} 1 & \text{if } \hat{p} \in B, \\ -1 & \text{otherwise.} \end{cases}$$

This proved more stable than continuous rewards but lacked sensitivity to error magnitude.

3. *Customized Discrete Reward (final choice):* To distinguish between predictions that miss the bounding box by a small or large margin, and to encourage predictions inside the box to move closer

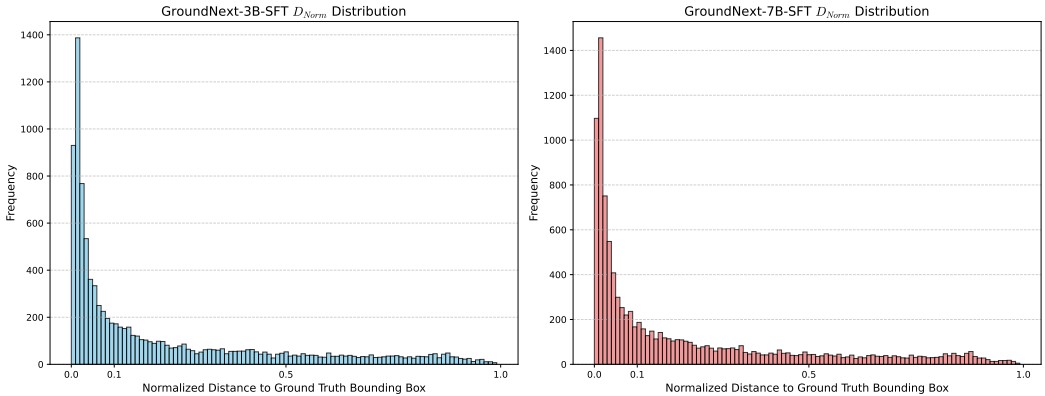

Figure 8: $\mathcal{D}_{norm}$ distribution of the errors made by GROUNDNEXT-3B (SFT) and GROUNDNEXT-7B (SFT) on the training set. 50% of the errors lie within $\mathcal{D}_{norm} < 0.1$, highlighting the motivation of our reward function.

to the center, we designed a customized discrete reward based on a normalized signed distance. The reward function is defined as:

$$R_{score}(\hat{p}, B, I) = \begin{cases} -1.0 & \text{if } \mathcal{D}_{norm} < -0.5, \\ -0.5 & \text{if } -0.5 \leq \mathcal{D}_{norm} < -0.1, \\ -0.1 & \text{if } -0.1 \leq \mathcal{D}_{norm} < 0, \\ 0.1 & \text{if } 0 \leq \mathcal{D}_{norm} < 0.1, \\ 0.5 & \text{if } 0.1 \leq \mathcal{D}_{norm} < 0.5, \\ 1.0 & \text{if } \mathcal{D}_{norm} \geq 0.5. \end{cases}$$

The normalized distance is calculated as $\mathcal{D}_{norm} = \frac{\mathcal{D}(\hat{p}, B)}{\mathcal{D}_{ref}}$, where $\mathcal{D}(\hat{p}, B)$ is the signed distance between the predicted coordinate $\hat{p}$ and the ground-truth bounding box $B$ (with positive values denoting the interior). The reference distance $\mathcal{D}_{ref}$ adapts based on the prediction's location to ensure $\mathcal{D}_{norm} \in [-1, 1]$:

$$\mathcal{D}_{ref} = \begin{cases} 0.5 \times \text{diam}(B) & \text{if } \hat{p} \in B, \\ \mathcal{D}_{max}(B, I) & \text{otherwise.} \end{cases}$$

Here, we use half the bounding box diameter when $\hat{p} \in B$, as this represents the maximum possible distance a point inside $B$ can have from the boundary. Conversely, $\mathcal{D}_{max}(B, I)$ represents the maximum distance in the image context $I$ for exterior points.

In summary, we adopt RLOO with this shaped reward formulation, as it effectively balances penalties for misses with incentives for precise centering. Our level-wise reward is motivated by the large proportion of predicted points that miss the bounding box by only a small margin. We highlight this characteristic in Figure 8, where we compute $\mathcal{D}_{norm}$ for 10K errors made by GROUNDNEXT-3B (SFT) and GROUNDNEXT-7B (SFT) on the training set they were trained on. This figure shows the imbalance in the error distance of the predicted points and the prevalence of "near misses", which directly motivates our choice of reward function.

To better demonstrate how the rewards behave for different predicted points, Figure 9 shows a screenshot from FreeCAD where the ground-truth bounding box encloses the "sketch in progress". In this toy example, we illustrate six predicted points, each with a corresponding $\mathcal{D}_{norm}$ value that falls into one of the ranges defined by our reward function. As a result, each point receives a different reward. Higher rewards correspond to predicted points that are closer to the center of the bounding box.

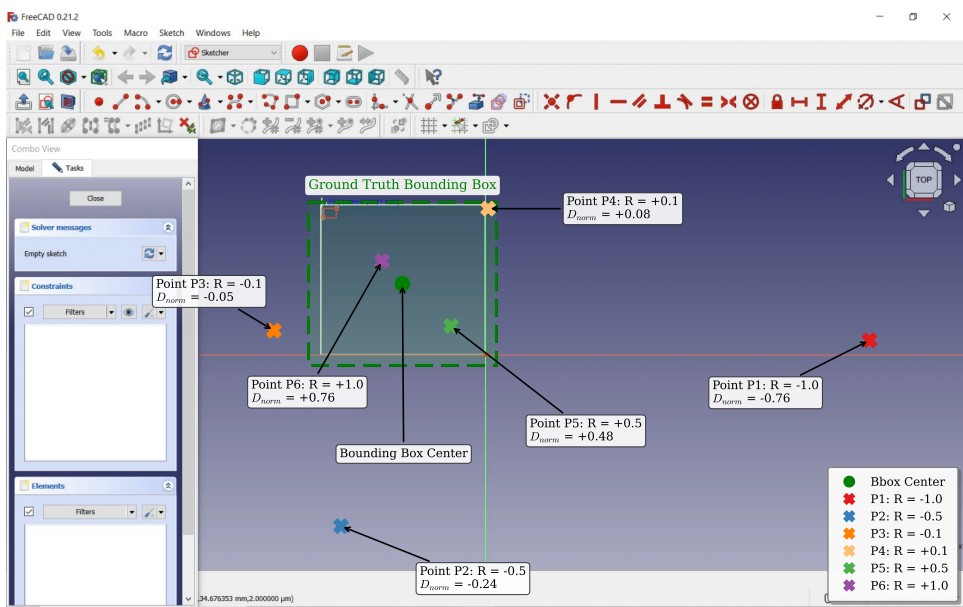

Figure 9: Rewards of 6 predicted points with respect to the ground truth bounding box in a screenshot of FreeCAD.

## C.5 RL Design Choices and Hyperparameters

### C.5.1 Discrete Reward Design

We investigate the impact of reward granularity on model performance during the RLOO stage. As shown in Table 8, we compared reward configurations ranging from binary feedback to 8-level quantization using a preliminary 3B checkpoint. We observe that the 6-level design achieves the best performance on OSWorld-G (63.1%) and yields the highest average score (65.2%) across all granularity settings. Based on these results, we selected the 6-level configuration (formulated as $\{-1.0, -0.5, -0.1, 0.1, 0.5, 1.0\}$) for all subsequent experiments.

Table 8: Performance of different reward designs for the RLOO stage using a preliminary 3B checkpoint. The "Levels" column indicates the number of discrete reward values used within the range $[-1, 1]$.

| Reward Granularity | MMBench-GUI | OSWorld-G | ScreenSpot-Pro | Avg |
|---|---|---|---|---|
| 8 Levels | 80.6 | 61.5 | 52.3 | 64.8 |
| 6 Levels | **81.0** | **63.1** | 51.6 | **65.2** |
| 4 Levels | 80.4 | 62.6 | **52.6** | 65.2 |
| 2 Levels | 81.0 | 62.1 | 52.4 | 65.1 |

### C.5.2 Impact of Group Number

We also ablated the group size $n$ (the number of generations per prompt) to balance performance with training efficiency. This study was conducted using a preliminary GROUNDNEXT-3B-SFT checkpoint trained via GRPO on a subset of 5.1K RL data points. As shown in Table 9, increasing the group size to $n = 32$ yields the best overall performance; however, the marginal gains do not justify the significant increase in computational cost and training time. While $n = 4$ slightly outperforms $n = 8$, we consider such a small group size potentially unstable for gradient estimation.

Consequently, we choose to $n = 8$ for our final experiments, a choice consistent with settings used in recent prior works (Liu et al., 2025a; Tang et al., 2025).

Table 9: Ablation of the group number $n$ using a preliminary 3B checkpoint with GRPO (5K samples). All scores are reported as percentages.

| $n$ | ScreenSpot-Pro | ScreenSpot-v2 | OSWorld-G | UI-Vision | Avg |
|---|---|---|---|---|---|
| 4 | 48.1 | 88.1 | **59.4** | 58.11 | 63.4 |
| 8 | 48.6 | 87.7 | 59.0 | 57.8 | 63.3 |
| 32 | **49.5** | **88.2** | 58.3 | **58.2** | **63.5** |

### C.5.3 RLOO vs. GRPO

Further, we conducted an ablation on RLOO vs. GRPO. While GRPO is widely used, we selected RLOO for its simplicity and its successful application in related work such as InfiGUI-G1 (Liu et al., 2025a). To validate this choice, we conducted a minimal comparison using an early SFT checkpoint (Table 10). We observed that RLOO achieves performance parity with GRPO, yielding a slightly higher average score (65.7% vs. 65.3%) and stronger results on OSWorld-G. We emphasize that we do not claim RLOO is inherently superior to GRPO; rather, these results indicate that RLOO was a reliable and robust configuration for our dataset-centric experiments.

Table 10: Comparison of RLOO and GRPO algorithms on an early SFT checkpoint. All scores are reported as percentages.

| Algorithm | ScreenSpot-Pro | ScreenSpot-v2 | OSWorld-G | Avg |
|---|---|---|---|---|
| GRPO | **49.5** | 88.2 | 58.3 | 65.3 |
| RLOO | 49.3 | **88.3** | **59.4** | **65.7** |

## D    EVALUATION

### D.1    SCREENSPOTPRO RESULTS

Table 11 summarises the results for different models on ScreenSpot-Pro (Li et al., 2025).

### D.2    OSWORLD-G RESULTS

Table 12 summarises the results for different models on OSWorld-G (Xie et al., 2025).

### D.3    MMBENCH-GUI RESULTS

Table 13 summarises the results for different models on MMBench-GUI (Wang et al., 2025b).

### D.4    SCREENSPOT-V2 RESULTS

Table 14 summarises the results for different models on ScreenSpot-v2 (Cheng et al., 2024).

### D.5    UI-VISION RESULTS

Table 15 summarises the results for different models on UI-Vision (Nayak et al., 2025).

Table 11: Performance of different models on SSPro across categories (CAD, Dev, Creative, Scientific, Office, OS). Text and Icon refer to different input types.

| Model | CAD | | Dev | | Creative | | Scientific | | Office | | OS | | Avg. | | |
|---|---|---|---|---|---|---|---|---|---|---|---|---|---|---|---|
| | Text | Icon | Text | Icon | Text | Icon | Text | Icon | Text | Icon | Text | Icon | Text | Icon | Avg. |
| GPT-4o | 2.0 | 0.0 | 1.3 | 0.0 | 1.0 | 0.0 | 2.1 | 0.0 | 1.1 | 0.0 | 0.0 | 0.0 | 1.3 | 0.0 | 0.8 |
| Claude Computer Use | 14.5 | 3.7 | 22.0 | 3.9 | 25.9 | 3.4 | 33.9 | 15.8 | 30.1 | 16.3 | 11.0 | 4.5 | 23.4 | 7.1 | 17.1 |
| Qwen2.5-VL-3B | 9.1 | 7.3 | 22.1 | 1.4 | 26.8 | 2.1 | 38.2 | 7.3 | 33.9 | 15.1 | 10.3 | 1.1 | 23.6 | 3.8 | 16.1 |
| Qwen2.5-VL-7B | 16.8 | 1.6 | 46.8 | 4.1 | 35.9 | 7.7 | 49.3 | 7.3 | 52.5 | 20.8 | 37.4 | 6.7 | 38.9 | 7.1 | 26.8 |
| FOCUS-2B | 7.6 | 3.1 | 22.8 | 1.7 | 23.7 | 1.7 | 25.0 | 7.1 | 23.2 | 7.7 | 17.8 | 2.5 | 19.8 | 3.9 | 13.3 |
| ShowUI-2B | 2.5 | 0.0 | 16.9 | 1.4 | 9.1 | 0.0 | 13.2 | 7.3 | 15.3 | 7.5 | 10.3 | 2.2 | 10.8 | 2.6 | 7.7 |
| UI-TARS-2B | 15.8 | 1.2 | 51.9 | 2.8 | 47.5 | 9.7 | 57.6 | 14.5 | 60.5 | 13.2 | 38.3 | 7.9 | 45.2 | 8.1 | 31.1 |
| JEDI-3B | 27.4 | 9.4 | 61.0 | 13.8 | 53.5 | 8.4 | 54.2 | 18.2 | 64.4 | 32.1 | 38.3 | 9.0 | 49.8 | 13.7 | 36.1 |
| SeeClick-9.6B | 2.5 | 0.0 | 0.6 | 0.0 | 1.0 | 0.0 | 3.5 | 0.0 | 1.1 | 0.0 | 2.8 | 0.0 | 1.8 | 0.0 | 1.1 |
| Aria-UI | 7.6 | 1.6 | 16.2 | 0.0 | 23.7 | 2.1 | 27.1 | 6.4 | 20.3 | 1.9 | 4.7 | 0.0 | 17.1 | 2.0 | 11.3 |
| OS-Atlas-7B | 12.2 | 4.7 | 33.1 | 1.4 | 28.8 | 2.8 | 37.5 | 7.3 | 33.9 | 5.7 | 27.1 | 4.5 | 28.1 | 4.0 | 18.9 |
| UGround-7B | 14.2 | 1.6 | 26.6 | 2.1 | 27.3 | 2.8 | 31.9 | 2.7 | 31.6 | 11.3 | 17.8 | 0.0 | 25.0 | 2.8 | 16.5 |
| UI-TARS-7B | 17.8 | 4.7 | 47.4 | 4.1 | 42.9 | 6.3 | 56.9 | 17.3 | 50.3 | 17.0 | 21.5 | 5.6 | 39.6 | 8.4 | 27.7 |
| JEDI-7B | 38.0 | 14.1 | 42.9 | 11.0 | 50.0 | 11.9 | 72.9 | 25.5 | 75.1 | 47.2 | 33.6 | 16.9 | 52.6 | 18.2 | 39.5 |
| GUI-Actor-7B | – | – | – | – | – | – | – | – | – | – | – | – | – | – | 44.6 |
| OpenCUA-7B | – | – | – | – | – | – | – | – | – | – | – | – | – | – | 50.0 |
| CogAgent-18B | 7.1 | 3.1 | 14.9 | 0.7 | 9.6 | 0.0 | 22.2 | 1.8 | 13.0 | 0.0 | 5.6 | 0.0 | 12.0 | 0.8 | 7.7 |
| UI-TARS-72B | 18.8 | 12.5 | 62.9 | 17.2 | 57.1 | 15.4 | 64.6 | 20.9 | 63.3 | 26.4 | 42.1 | 15.7 | 50.9 | 17.6 | 38.1 |
| UI-R1-3B | 11.2 | 6.3 | 22.7 | 4.1 | 27.3 | 3.5 | 42.4 | 11.8 | 32.2 | 11.3 | 13.1 | 4.5 | 24.9 | 6.4 | 17.8 |
| UI-R1-E-3B | 37.1 | 12.5 | 46.1 | 6.9 | 41.9 | 4.2 | 56.9 | 21.8 | 65.0 | 26.4 | 32.7 | 10.1 | – | – | 33.5 |
| GUI-R1-3B | 26.4 | 7.8 | 33.8 | 4.8 | 40.9 | 5.6 | 61.8 | 17.3 | 53.6 | 17.0 | 28.1 | 5.6 | – | – | – |
| InfiGUI-R1-3B | 33.0 | 14.1 | 51.3 | 12.4 | 44.9 | 7.0 | 58.3 | 20.0 | 65.5 | 28.3 | 43.9 | 12.4 | 49.1 | 14.1 | 35.7 |
| GUI-G1-3B | 39.6 | 9.4 | 50.7 | 10.3 | 36.6 | 11.9 | 61.8 | 30.0 | 67.2 | 32.1 | 23.5 | 10.6 | 49.5 | 16.8 | 37.1 |
| SE-GUI-3B | 38.1 | 12.5 | 55.8 | 7.6 | 47.0 | 4.9 | 61.8 | 16.4 | 59.9 | 24.5 | 40.2 | 12.4 | 50.4 | 11.8 | 35.9 |
| InfiGUI-G1-3B | 50.8 | 25.0 | 64.9 | 20.0 | 51.5 | 16.8 | 68.8 | 32.7 | 70.6 | 32.1 | 49.5 | 15.7 | – | – | 45.2 |
| GUI-R1-7B | 23.9 | 6.3 | 49.4 | 4.8 | 38.9 | 8.4 | 55.6 | 11.8 | 58.7 | 26.4 | 42.1 | 16.9 | – | – | – |
| SE-GUI-7B | 51.3 | 42.2 | 68.2 | 19.3 | 57.6 | 9.1 | 75.0 | 28.2 | 78.5 | 43.4 | 49.5 | 25.8 | 63.5 | 21.0 | 47.3 |
| Phi-Ground-7B-16C-DPO | **70.8** | 16.7 | 56.6 | 13.3 | 26.9 | 17.2 | 58.0 | 29.1 | 76.4 | 44.0 | 55.1 | 25.8 | 56.4 | 21.8 | 43.2 |
| GUI-G$^2$-7B | 55.8 | 12.5 | 68.8 | 17.2 | 57.1 | 15.4 | 77.1 | 24.5 | 74.0 | 32.7 | 57.9 | 21.3 | 64.7 | 19.6 | 47.5 |
| GTA1-7B | 66.9 | 20.7 | 62.6 | 18.2 | 53.3 | 17.2 | 31.8 | 76.4 | 82.5 | 50.9 | 48.6 | 25.9 | 65.5 | 25.2 | 50.1 |
| InfiGUI-G1-7B | 57.4 | 23.4 | **74.7** | 24.1 | **64.6** | 15.4 | **80.6** | 31.8 | 75.7 | 39.6 | **57.0** | 29.2 | **68.4** | 25.2 | 51.9 |
| **Our Models** | | | | | | | | | | | | | | | |
| **GROUNDNEXT-3B (SFT)** | 50.3 | 26.6 | 65.6 | 36.6 | 48.5 | 22.4 | 66.0 | 38.2 | 76.3 | 54.7 | 41.1 | 28.1 | 58.3 | 32.8 | 48.6 |
| **GROUNDNEXT-3B (RL)** | 55.3 | 32.8 | 65.6 | 36.6 | 50.0 | 24.5 | 66.0 | 37.3 | 74.6 | 50.9 | 45.8 | 29.2 | 59.9 | 33.6 | 49.8 |
| **GROUNDNEXT-7B (SFT)** | 46.2 | 32.8 | 68.2 | 38.6 | 54.5 | 20.3 | 70.8 | 37.3 | **76.8** | 49.1 | 45.8 | 33.7 | 59.9 | 33.6 | 50.2 |
| **GROUNDNEXT-7B (RL)** | 50.2 | **34.3** | 73.4 | **40.0** | 59.6 | **23.8** | 70.1 | **42.7** | 74.6 | **54.7** | 53.3 | 30.3 | 60.5 | **33.6** | **52.9** |

Table 12: Performance comparison of models on OSWORLD-G across multiple capability dimensions.

| Model | Text Matching | Element Recognition | Layout Understanding | Fine-grained Manipulation | Refusal | Overall |
|---|---|---|---|---|---|---|
| OS-Atlas-7B | 44.1 | 29.4 | 35.2 | 16.8 | 7.4 | 27.7 |
| UGround-V1-7B | 51.3 | 40.3 | 43.5 | 24.8 | 0.0 | 36.4 |
| Aguvis-7B | 55.9 | 41.2 | 43.9 | 28.2 | 0.0 | 38.7 |
| UI-TARS-7B | 60.2 | 51.8 | 54.9 | 35.6 | 0.0 | 47.5 |
| Seed1.5-VL | 73.9 | 66.7 | 69.6 | 47.0 | 18.5 | 62.9 |
| UI-TARS-72B | 69.4 | 60.6 | 62.9 | 45.6 | 0.0 | 57.1 |
| Gemini-2.5-Pro | 59.8 | 45.5 | 49.0 | 33.6 | **38.9** | 45.2 |
| Operator | 51.3 | 42.4 | 46.6 | 31.5 | 0.0 | 40.6 |
| Qwen2.5-VL-3B | 41.4 | 28.8 | 34.8 | 13.4 | 0.0 | 27.3 |
| Qwen2.5-VL-7B | 45.6 | 32.7 | 41.9 | 18.1 | 0.0 | 31.4 |
| Qwen2.5-VL-32B | 63.2 | 47.3 | 49.0 | 36.9 | 0.0 | 46.5 |
| JEDI-3B | 67.4 | 53.0 | 53.8 | 44.3 | 7.4 | 50.9 |
| JEDI-7B | 65.9 | 55.5 | 57.7 | 46.9 | 7.4 | 54.1 |
| InfiGUI-G1-3B | 65.5 | 53.0 | 56.1 | 34.2 | 0.0 | 49.6 |
| InfiGUI-G1-7B | 72.0 | 63.6 | 66.8 | 46.3 | 0.0 | 59.9 |
| GTA-1-7B | 63.2 | **82.1** | **74.2** | **70.5** | 0.0 | **67.7** |
| **Our Models** | | | | | | |
| **GROUNDNEXT-3B (SFT)** | 67.4 | 68.8 | 68.4 | 43.0 | 0.0 | 62.2 |
| **GROUNDNEXT-3B (RL)** | 70.9 | 71.2 | 70.8 | 43.6 | 0.0 | 64.2 |
| **GROUNDNEXT-7B (SFT)** | 72.4 | 73.3 | 73.1 | 53.7 | 0.0 | 67.2 |
| **GROUNDNEXT-7B (RL)** | **74.3** | 73.9 | 73.5 | 51.7 | 0.0 | **67.7** |

Table 13: MMBench-GUI: Cross-platform performance of models across Windows, MacOS, Linux, iOS, Android, and Web.

| Model | Windows | | MacOS | | Linux | | iOS | | Android | | Web | | Avg |
|---|---|---|---|---|---|---|---|---|---|---|---|---|---|
| | Basic | Adv. | Basic | Adv. | Basic | Adv. | Basic | Adv. | Basic | Adv. | Basic | Adv. | |
| GPT-4o | 1.5 | 1.1 | 8.7 | 4.3 | 1.1 | 1.0 | 5.1 | 3.3 | 2.5 | 1.4 | 3.2 | 2.9 | 2.9 |
| Claude-3.7 | 1.5 | 0.7 | 12.5 | 7.5 | 1.1 | 0.0 | 13.7 | 10.6 | 1.4 | 1.4 | 3.2 | 2.3 | 4.7 |
| Qwen-Max-VL | 43.9 | 36.8 | 58.8 | 56.1 | 53.9 | 30.1 | 77.4 | 59.1 | 79.5 | 70.1 | 74.8 | 58.8 | 58.0 |
| ShowUI-2B | 9.2 | 4.4 | 24.1 | 10.4 | 25.1 | 11.7 | 29.0 | 19.7 | 17.4 | 8.7 | 22.9 | 12.7 | 16.0 |
| Qwen2.5-VL-7B | 31.4 | 16.5 | 31.3 | 22.0 | 21.5 | 10.2 | 66.6 | 55.2 | 35.1 | 35.2 | 40.3 | 32.5 | 33.9 |
| Qwen2.5-VL-72B | 55.7 | 33.8 | 49.9 | 30.1 | 40.3 | 20.9 | 56.1 | 28.2 | 55.6 | 25.4 | 68.4 | 45.8 | 41.8 |
| OS-Atlas-Base-7B | 36.9 | 18.8 | 44.4 | 21.7 | 31.4 | 13.3 | 74.8 | 48.8 | 69.6 | 46.8 | 61.3 | 35.4 | 41.4 |
| Aguvis-7B-720P | 37.3 | 21.7 | 48.1 | 33.3 | 33.5 | 25.0 | 67.5 | 65.2 | 61.0 | 51.0 | 61.6 | 45.5 | 45.7 |
| UI-TARS-1.5-7B | 68.3 | 39.0 | 69.0 | 44.5 | 64.4 | 37.8 | 88.5 | 69.4 | 90.5 | 69.3 | 81.0 | 56.5 | 64.3 |
| UI-TARS-72B-DPO | 78.6 | 51.8 | 80.3 | 62.7 | 68.6 | 51.5 | 90.8 | 81.2 | 93.0 | 80.0 | 88.1 | 68.5 | 74.3 |
| UGround-V1-7B | 66.8 | 39.0 | 71.3 | 48.6 | 56.5 | 31.1 | 92.7 | 70.9 | 93.5 | 71.0 | 88.7 | 64.6 | 65.7 |
| InternVL3-72B | 70.1 | 42.6 | 75.7 | 52.3 | 59.2 | 41.3 | 93.6 | 80.6 | 92.7 | 78.6 | 90.7 | 65.9 | 72.2 |
| Naive RLVR-3B | 68.6 | 44.5 | 78.6 | 50.0 | 61.3 | 39.3 | 92.4 | 76.4 | 91.3 | 76.1 | 87.4 | 63.0 | 70.9 |
| Naive RLVR-7B | 79.3 | 58.1 | 82.3 | 62.7 | 64.4 | 44.9 | 94.9 | 89.1 | 95.5 | 84.2 | 92.9 | 79.5 | 79.3 |
| InfiGUI-G1-3B | 74.2 | 47.1 | 78.8 | 55.2 | 65.4 | 41.8 | 95.2 | 78.8 | 92.1 | 78.0 | 89.7 | 64.3 | 73.4 |
| InfiGUI-G1-7B | 82.7 | 61.8 | 83.8 | 63.9 | 72.3 | 52.0 | 94.9 | 89.4 | 95.2 | 85.6 | 93.5 | 76.3 | 80.8 |
| **Our Models** | | | | | | | | | | | | | |
| GROUNDNEXT-3B (SFT) | 81.5 | 50.7 | 85.8 | 64.2 | 73.8 | 53.6 | 93.0 | 77.0 | 90.4 | 73.8 | 88.1 | 59.7 | 75.5 |
| GROUNDNEXT-3B (RL) | 80.4 | 52.6 | 87.2 | 64.5 | 70.7 | 57.1 | 94.9 | 78.5 | 91.9 | 78.0 | 90.6 | 64.3 | 77.1 |
| GROUNDNEXT-7B (SFT) | 83.8 | 60.7 | 86.7 | 69.9 | 75.4 | 61.2 | 94.3 | 83.3 | 94.9 | 79.4 | 91.0 | 70.5 | 80.4 |
| GROUNDNEXT-7B (RL) | 81.5 | 60.7 | 87.8 | 73.1 | 75.4 | 59.2 | 95.2 | 86.1 | 95.5 | 80.3 | 90.97 | 72.7 | 81.1 |

Table 14: ScreenSpot-V2: Cross-platform breakdown by device and modality. "Icon/Widget" indicates icon- or widget-based queries. "Avg." is across all devices and modalities.

| Model | Mobile | | Desktop | | Web | | Avg. |
|---|---|---|---|---|---|---|---|
| | Text | Icon/Widget | Text | Icon/Widget | Text | Icon/Widget | |
| SeeClick | 78.4 | 50.7 | 70.1 | 29.3 | 55.2 | 32.5 | 55.1 |
| OS-Atlas-Base-7B | 95.2 | 75.8 | 90.7 | 63.6 | 90.6 | 77.3 | 85.1 |
| UI-TARS-7B | 96.9 | 89.1 | 95.4 | 85.0 | 93.6 | 85.2 | 91.6 |
| UI-TARS-72B | 94.8 | 86.3 | 91.2 | 87.9 | 91.5 | 87.7 | 90.3 |
| Qwen2.5-VL-3B | 93.4 | 73.5 | 88.1 | 58.6 | 88.0 | 71.4 | 80.9 |
| Qwen2.5-VL-7B | 97.6 | 87.2 | 90.2 | 74.2 | 93.2 | 81.3 | 88.8 |
| Qwen2.5-VL-32B | 97.9 | 88.2 | 98.5 | 79.3 | 91.2 | 86.2 | 91.3 |
| InfiGUI-G1-3B | 99.3 | 88.2 | 94.8 | 82.9 | 94.9 | 80.3 | 91.1 |
| InfiGUI-G1-7B | 99.0 | 91.9 | 94.3 | 82.1 | 97.9 | 89.2 | 93.5 |
| **Our Models** | | | | | | | |
| GROUNDNEXT-3B (SFT) | 95.2 | 80.6 | 93.8 | 84.3 | 87.6 | 78.8 | 87.3 |
| GROUNDNEXT-3B (RL) | 94.8 | 96.4 | 93.9 | 87.1 | 90.6 | 79.3 | 88.5 |
| GROUNDNEXT-7B (SFT) | 97.2 | 84.8 | 94.3 | 90.0 | 91.5 | 74.9 | 89.3 |
| GROUNDNEXT-7B (RL) | 96.6 | 88.2 | 95.4 | 87.9 | 94.9 | 75.9 | 90.4 |

## D.6 DISCUSSION ON RL GAINS

We hypothesize that the observed "limited" improvement is not a failure of the RL step, but rather a finding that highlights the interaction between strong SFT baselines and RL gains. We also demonstrate that GROUNDCUA is an effective dataset for RL fine-tuning. We detail our analysis below.

### D.6.1 HIGH-QUALITY SFT CREATES A "STRONG CEILING"

We hypothesize that stronger GUI models contain fewer actionable errors after the SFT stage, resulting in lower marginal benefits from subsequent RL fine-tuning, especially when the RL data is drawn from the same distribution as the SFT data. This is supported by Figure 3, where we observe

Table 15: UI-Vision: Performance grouped by category (Edu., Browser, Dev., Prod., Creative, Entert.) and by setting (Basic, Functional, Spatial).

| Model | Grouped by Setting | | | Overall |
|---|---|---|---|---|
| | Basic | Func. | Spatial | |
| GPT-4o | 1.6 | 1.5 | 1.0 | 1.4 |
| Claude-3.7-Sonnet | 9.5 | 7.7 | 7.6 | 8.3 |
| Qwen-2.5VL-7B | 1.2 | 0.8 | 0.5 | 0.9 |
| InternVL2.5-8B | 2.5 | 2.8 | 1.0 | 2.1 |
| MiniCPM-V-8B | 7.1 | 5.3 | 1.5 | 4.3 |
| SeeClick-9.6B | 9.4 | 4.7 | 2.1 | 5.4 |
| ShowUI-2B | 8.1 | 7.7 | 2.1 | 5.9 |
| CogAgent-9B | 12.0 | 12.2 | 2.6 | 8.9 |
| OSAtlas-7B | 12.2 | 11.2 | 3.7 | 9.0 |
| AriaUI-25.3B | 12.2 | 14.0 | 4.0 | 10.1 |
| UGround-v1-7B | 15.4 | 17.1 | 6.3 | 12.9 |
| UGround-v1-72B | 27.9 | 26.7 | 14.9 | 23.2 |
| Aguvis-7B | 17.8 | 18.3 | 5.1 | 13.7 |
| UI-TARS-7B | 20.1 | 24.3 | 8.4 | 17.6 |
| UI-TARS-72B | 31.4 | 30.5 | 14.7 | 25.5 |
| InfiGUI-G1-3B | 31.2 | 28.0 | 8.2 | 22.0 |
| InfiGUI-G1-7B | 36.2 | 31.9 | 11.5 | 26.1 |
| **Our Models** | | | | |
| GROUNDNEXT-3B (SFT) | 70.9 | 59.8 | 45.1 | **58.2** |
| GROUNDNEXT-3B (RL) | 72.9 | 63.9 | 50.6 | **62.1** |
| GROUNDNEXT-7B (SFT) | 67.1 | 60.0 | 49.9 | **58.7** |
| GROUNDNEXT-7B (RL) | 70.1 | 62.0 | 49.9 | **60.3** |

that RL provides significantly larger gains for SFT models trained with other datasets compared to GROUNDCUA. We attribute this to the fact that GROUNDCUA yields a much stronger initial model; consequently, the RL stage serves as a minor refinement rather than a primary performance driver in our current setting.

Table 16: Performance comparison between the base model (UI-Tars-1.5-7B) and the RL-tuned model (GTA-1-7B). Note that the results for UI-TARS-1.5-7B are reported using our own evaluation setup and differs from (Yang et al., 2025).

| Benchmark | UI-Tars-1.5-7B | GTA-1-7B | Improvement |
|---|---|---|---|
| **SS-Pro** | 47.9 | 50.1 | +2.2% |
| **OSW-G** | 64.2 | 67.7 | +3.5% |
| **MMB-GUI** | 75.4 | 79.4 | +4.0% |
| **SSv2** | 90.3 | 92.4 | +2.1% |
| **UI-V** | 20.8 | 25.7 | +4.9% |

We also see this trend in a related work, GTA1 (Yang et al., 2025). GTA1-7B initializes its training from UI-TARS-1.5-7B, which is a powerful GUI grounding model. We observe that the average improvement across five benchmarks is 3.3% (see Table 16). This demonstrates that, while RL provides a consistent lift, the gains are moderate, not universally massive. The limited marginal return observed in GROUNDNEXT is consistent with the observations for GTA-1-7B.

**Important Note:** We emphasize that we **do not make a general claim that stronger GUI models cannot be effectively RL-tuned**. Our results merely provide evidence supporting the hypothesis that the marginal return is lower when a robust, high-quality SFT initialization is used, particularly

for models in the 3–7B size range under the reward formulations we employ. A more detailed study spanning different architectures, various reward functions, different model sizes, and deeper RL fine-tuning is required to fully understand these interactions. This comprehensive investigation is beyond the scope of our current paper, but our results provide the initial evidence for an interesting phenomenon.

### D.6.2 GROUNDCUA IS A GREAT SOURCE FOR RL FINE-TUNING

Table 17: RL ablation study: Performance of the Qwen2.5-VL-3B baseline trained with 10k RL samples from different datasets.

| Dataset | SSPro | OSWorld-G | SSv2 | MMBench | UI-V |
|---|---|---|---|---|---|
| **Baseline** | 29.0 | 37.4 | 81.8 | 60.8 | 6.3 |
| **Aguvis** | 31.2 | 45.6 | 86.01 | 67.01 | 14.7 |
| **OSAtlas** | 30.4 | 46.4 | 62.0 | 67.7 | 14.1 |
| **UGround** | 33.6 | 43.8 | **89.0** | 68.8 | 16.7 |
| **GroundCUA** | **36.8** | **48.8** | 88.9 | **70.5** | **19.2** |
| **Imp. over baseline** | +7.8% | +11.4% | +7.1% | +9.7% | +12.9% |

We clarify that the limited marginal gain observed in the final GROUNDNEXT models is not a flaw of the GROUNDCUA dataset itself. We validate this through a controlled experiment where we sampled 10k data points from GROUNDCUA and three competing datasets with available bounding boxes: Aguvis, OS-Atlas, and UGround. We trained a Qwen2.5-VL-3B-Instruct baseline using the hyperparameters described in Section 4.1 and Appendix C.4, with two modifications: we adopted the simpler binary (0/1) reward formulation described in the GTA paper and extended training to 2 epochs, as we observed rewards continuing to increase after the first epoch. We report the performance across various benchmarks in Table Table 17

We observe that GROUNDCUA provides substantial gains of 9.8% on average over the Qwen2.5-VL-3B baseline. Furthermore, it achieves an average gain of 1.9% over the next best baseline (and 2.5% if we exclude the SSv2 benchmark). This validates that the GROUNDCUA data is highly effective for RL. We attribute this to our human-annotated labels and bounding boxes (less noise) and the rich diversity of platforms covered by our dataset.

The strength of GROUNDCUA lies in its platform diversity (87 applications across 12 categories) and dense annotations, which offer a huge variety of UI elements for agents to learn. RL training on a very large scale (e.g., 700k samples in our case) is computationally expensive, especially in resource-constrained settings (e.g., we used 8 H100 GPUs for our experiments). Hence, we believe the diversity of GROUNDCUA can be effectively exploited through a careful combination of SFT, which teaches new knowledge, and RL, which helps with generalization. Future research could focus on optimizing this SFT/RL mix, which has shown promise in works like Ma et al. (2025). By releasing our data, we provide the necessary resource to explore this path.

### D.7 SFT SCALING BEHAVIOR

To study how performance scales with additional supervised data, we trained three versions of the Qwen2.5-VL-Instruct-3B model using subsets of 100k, 350k, and 700k instructions sampled from GROUNDCUA. All runs used identical hyperparameters. The results are reported in Table 18.

We observe steady gains across almost all benchmarks as the amount of training data increases. The improvements indicate that the training setup remains stable across scales and that the additional samples provide useful signal. Since GROUNDCUA undergoes rigorous deduplication, larger subsets introduce new visual and semantic variety rather than repeated patterns, which directly strengthens grounding performance.

The most pronounced improvements appear on UI-Vision, where performance increases from 29.8 at 100k to 58.2 at 700k. UI-Vision is closely aligned with the layouts and element styles present

Table 18: Scaling ablation on GROUNDCUA. Performance improves steadily as more SFT data is used.

| Size | SS-Pro | OS-World-G | MMBench-GUI | UI-Vision | SS-v2 | Average |
|------|--------|-----------|-------------|-----------|-------|---------|
| 100k | 41.7 | 54.6 | 72.5 | 29.8 | 87.4 | 57.2 |
| 350k | 44.5 | 57.3 | 72.9 | 39.5 | 87.0 | 60.2 |
| 700k | 48.6 | 62.2 | 75.5 | 58.2 | 87.3 | 66.4 |

in GROUNDCUA, allowing the model to leverage broader coverage as more data is included. OSWorld-G and ScreenSpot-Pro show similar positive trends, reflecting consistent benefits in dense desktop scenarios.

Overall, the scaling results show that high-quality, non-redundant supervision continues to improve grounding accuracy throughout this data range, suggesting room for further gains with larger curated subsets.

# E GROUNDNEXT ERROR ANALYSIS

Prompt: join a twitch server

Prompt: view ore details

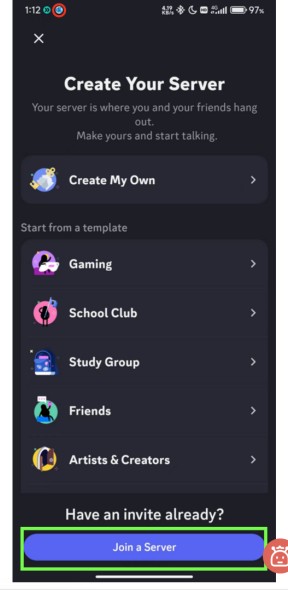
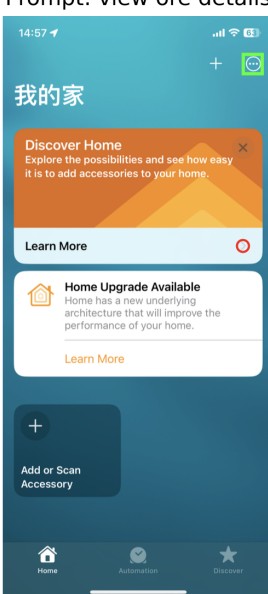

Figure 10: Errors made by GROUNDNEXT on mobile devices. Examples are chosen from the SSv2 benchmark.

In this section, we examine the errors made by GROUNDNEXTs and categorise them into 4 broad categories:

**Limited Domain Knowledge (Generalization to Web/Mobile):** We observe that some errors stem from limited knowledge of web and mobile platforms, as GROUNDCUA predominantly covers desktop software applications. This is most prevalent when generalizing to out-of-domain platforms. While GROUNDNEXT performs competitively on Mobile and Web benchmarks, errors often arise due to distribution shifts. Mobile interfaces, for instance, utilize vastly different aspect ratios, resolutions, and distinct UI patterns that are absent in our desktop-centric training data. Figure 10 illustrates examples where GROUNDNEXT fails on relatively simple queries for the mobile platform,

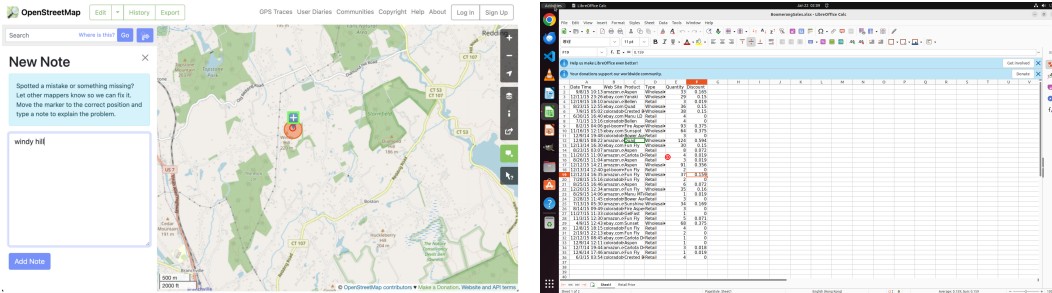

Figure 11: (Left) Example of a near miss for the prompt "Click on the marker already added to the map." (Right) Example from OSWorld-G where the model clicks the wrong cell for the prompt "Move your mouse to the cell in the 3rd column and 12th row (this cell is labeled as C12), then press the left mouse button."

such as 'Join a twitch server.' We attribute these errors to a combination of factors, primarily the domain shift inherent in mobile screenshots compared to our desktop training data, as well as a lack of specific application knowledge, which we discuss in greater detail below.

**Localization Precision (Near-Misses):** We analyzed the magnitude of grounding errors and found that many "failures" are actually correct semantics with imperfect localization. As shown in Figure 8, over 50% of the errors have a relative distance of less than 10% from the ground truth bounding box. This suggests the model successfully identifies the correct target region but occasionally lacks pixel-perfect precision. We visualize a "near-miss" example in Figure Figure 11 (left).

**Application-Specific Semantics:** We observe errors when the model encounters specialized terminology or icons in unseen software applications. For example, in our analysis of Platform VMWare in ScreenSpot-Pro (which is not present in our training data), the model struggles with niche tools that require specific software knowledge to identify (eg, "restart from CD", "snapshot details"), whereas it remains robust on generic UI elements like "Refresh", "font size".

**Spatial Reasoning Limitations:** Despite the inclusion of spatial data in our instruction mix ($\approx$ 13%), the model shows a notable performance gap when handling complex relative instructions. This is quantified in the UI-Vision benchmark, where performance drops from $\approx 70.1\%$ on the "Basic" category to $\approx 49.9\%$ on the "Spatial" category. We suspect that while the current data helps, solving this fully may require a base model with stronger inherent spatial reasoning capabilities or a higher ratio of spatial instruction tuning. In Figure 11 (right), we show an error in localising the correct cell in the Libre Office Calc platform (from the OSWorld-G benchmark)

## F    LIMITATIONS

While our work makes significant progress in desktop GUI grounding, there are a few key limitations. Although it covers 87 applications across 6 categories, the dataset may not fully represent the diversity of desktop software, as it is biased toward commonly used applications. Our keyframe-based annotations capture static UI states but miss dynamic elements like animations and real-time updates. While we've taken steps to ensure annotation consistency, human labeling at scale can still introduce some inconsistencies, and the time and cost of annotation limit scalability. Additionally, our evaluation focuses on benchmark accuracy, but real-world applications require robustness to changes in distribution, new app versions, and UI updates; issues that need further exploration. Finally, we do not perform end-to-end agentic testing for task completion, which remains an important area for future work.

## G    LLM USAGE

In our work, LLMs are used for the following aspects:

- Using an LLM to help with paper writing. We use GPT5 to help optimize language, correct grammar, and write LaTeX table code.

- Using an LLM as a research assistant. We use GPT5 to help search related works.

- Using LLMs in our methods and experiments. This is described in the paper.

