# OpenReview forum: "Grounding Computer Use Agents on Human Demonstrations"
_ICLR.cc/2026/Conference — ICLR 2026 Poster_

### Official Review · Reviewer_Xw9S · 2025-10-31

**Soundness:** 3
**Presentation:** 3
**Contribution:** 3
**Rating:** 6
**Confidence:** 4

**Summary:**

This paper introduces GROUNDCUA, a large-scale human-expert-annotated dataset for desktop UI grounding. The dataset is comprehensive, covering 56K screenshots from 87 applications with over 3.56M dense annotations. The authors also present GROUNDNEXT, a series of models (3B and 7B) trained on this dataset using a two-stage SFT-RL process. GROUNDNEXT achieves state-of-the-art performances on five grounding benchmarks, significantly outperforming models trained on datasets 10x larger. The paper argues for the superior value of high-quality, dense, expert-driven data over sheer data quantity for this task.

**Strengths:**

- The GROUNDCUA dataset. It collects grounding samples during human-driven task execution. The inclusion of complex and diverse desktop screen states directly addresses a major gap in existing resources.
- Sample efficiency. This is another valuable feature of the dataset. Training with significantly fewer samples, GROUNDNEXT can outperform models with much more samples.

**Weaknesses:**

- Although during data collection, synthetic tasks are written by human and conducted by human in real environments, only the grounding samples automatically extracted from each step is collected for training a grounding-only model. Can the authors elaborate more on the reasons and rationales behind such design?
- A highly relevant work on scaling desktop UI data and training a grounding model is lacking discussion in the paper: Aria-UI: Visual Grounding for GUI Instructions.
- Limited RL impact: 66.4->68.4 and 69.2->70.5. Considering that in GTA1, RL training brings very significant improvements, what's the underlying factors that lead to limited RL impact on GROUNDNEXT after SFT? Would it be relevant to the data? Also, it might be valuable to see how the GTA1-style heavy RL training for grounding would work with the GROUNDCUA dataset.
- Is there a timeline for open-sourcing the tasks and trajectories?

**Questions:**

Please see weaknesses.

---

> ### Author Response · Authors · 2025-11-24
> **Response to Reviewer Xw9S (1/3)**
>
> **Q1:** *"Although during data collection, [...] model. Can the authors elaborate more on the reasons and rationales behind such a design?"*
>
> **Response:** We thank the reviewer for this insightful question.
>
> First, a brief but important clarification on the reviewer's observation: the grounding samples themselves are not automatically extracted. The process of annotating all on-screen elements with bounding boxes and labels is performed manually by human experts. This expert-driven, manual-annotation process is a core part of our contribution, not an automated step.
>
> Our rationale for this "grounding-first" design is based on three main points:
> 1. **Grounding is the Primary Bottleneck:** Our work is motivated by the fact that inaccurate grounding is a primary failure mode for computer-use agents. Several works like [1], [2], [3] have pointed this out. As we state in our paper, even a flawless high-level plan will fail if the agent cannot accurately ground that instruction. Hence, we chose to focus on solving this foundational component robustly.
> 2. **Dense Supervision vs. Sparse Actions:** An action trajectory is sparse, as it typically involves only one element at a given step. We, however, wanted to solve the dense problem of understanding the entire visual context of the screen, which we hypothesize is necessary to perform actions reliably. We created a dataset that explicitly provides this dense supervision, which is why we had humans annotate every visible element in a screenshot, not just the single element that was the target of the action.
> 3. **Addressing Gaps in Desktop Data Collection:** As shown in Table 1, high-quality desktop data is less common than web and mobile data. This motivated us to specifically target desktop platforms. Furthermore, existing methods often rely on accessibility trees, which are frequently incomplete, or use automated BFS/DFS/MLLM-powered random walks, which may not capture the true distribution of UI states humans encounter. Our method of using human-designed tasks ensures our data (specifically the screenshots) is sampled from task-centric interaction states.
>
> We absolutely agree that the complete tasks and video trajectories are valuable. We leave the challenge of training end-to-end models that combine planning with grounding for future work. We will be releasing the tasks and full videos to enable precisely this line of research. We hope this answers the reviewer’s question. Please let us know if further clarification is needed.
>
> ---
>
> **Q2:** *"A highly relevant work on scaling desktop UI data and training a grounding model is lacking discussion in the paper: Aria-UI: Visual Grounding for GUI Instructions."*
>
>  **Response:** We thank the reviewer for highlighting Aria-UI. We agree it is a relevant work and have included it as a baseline in our evaluations (please refer to Tables 11 and 15).
>
> We acknowledge that a detailed comparison was missing from Section 2 (Related Work) and Table 1. We have revised the paper to include this and clarify the key differences in our dataset. As the reviewer notes, Aria-UI is a valuable dataset, though its data collection is predominantly focused on web elements, and its Desktop data is scraped using an MLLM exploring the OS and accessibility trees, as compared to the human demonstrations in our case. We have updated Table 1 to summarize the key distinctions:
>
> | Grounding Datasets | H | Desk | E | Desk-E | S | Res Range | EleArea | #AvgE | Perm? |
> | :--- | :---: | :---: | :---: | :---: | :---: | :---: | :---: | :---: | :---: |
> | AriaUI [4] | ✗ | ✓ | 4.1M | 150K | 295K | (1.3, 1.9) | — | 13.9 | ? |
> | **GROUNDCUA (ours)** | ✓ | ✓ | 3.56M | 3.56M | 55k | (0.4, 7.0) | 0.13% | 64.1 |  ✓
>
> Our GroundCUA dataset contributes a significantly larger-scale, human-annotated resource for Desktop environments. It features a greater diversity of screenshot resolutions and more densely annotated screenshots.

---

> ### Author Response · Authors · 2025-11-24
> **Response to Reviewer Xw9S (2/3)**
>
> **Q3:** *"Limited RL impact: 66.4->68.4 and 69.2->70.5. Considering that in GTA1, RL [...] data? Also, it might be valuable to see how the GTA1-style heavy RL training for grounding would work with the GROUNDCUA dataset."*
>
> **Response:** We thank the reviewer for raising this point regarding the limited RL impact, as it highlights a fascinating area of model behavior.
>
>
> ## **High-Quality SFT Creates a "Strong Ceiling"**
>
>
> We want to first address the marginal benefits obtained from RL in GroundNext models. We suspect that stronger GUI models, which leave fewer actionable errors, yield lower marginal benefits from further RL fine-tuning, especially when the RL data comes from the same distribution as the data used for SFT. Besides the observations reported in Figure 3, we also want to highlight similar patterns in GTA-1.
>
> We firstly note that GTA-1-7B initializes its training from UI-Tars-1.5-7B, which is a powerful GUI grounding model. When we analyze the observed gains across all five major benchmarks, the average improvement is moderate, supporting the idea that the marginal return for RL on top of a strong SFT baseline is inherently challenging. We provide a detailed comparison below:
>
> | Benchmark | UI-Tars-1.5-7B | GTA-1-7B | Improvement |
> | :--- | :--- | :--- | :--- |
> | **SS-Pro** | 47.9 | 50.1 | 2.2% |
> | **OSW-G** | 64.2 | 67.7 | 3.5% |
> | **MMB-GUI** | 75.4 | 79.4 | 4.0% |
> | **SSv2** | 90.3 | 92.4 | 2.1% |
> | **UI-V** | 20.8 | 25.7 | 4.9% |
>
> We observe that the average improvement across five benchmarks is 3.3%. This demonstrates that, while RL provides a consistent lift, the gains are moderate, not universally massive. The limited marginal return observed in GroundNext is consistent with the observations for GTA-1-7B.
>
> **Important Note:** We emphasize that we **do not make a general claim that stronger GUI models cannot be effectively RL-tuned.** Our results merely provide evidence supporting the hypothesis that the marginal return is lower when a robust, high-quality SFT initialization is used, particularly for models in the 3–7B size range under the reward formulations we employ.
>
> A more detailed study spanning different architectures, various reward functions, different model sizes, and deeper RL fine-tuning is required to fully understand these interactions. This comprehensive investigation is beyond the scope of our current paper, but our results provide the initial evidence for an interesting phenomenon.
>
> ## **GroundCUA is a great source for RL fine-tuning**
>
>
> We clarify that the limited marginal gain observed in the final GroundNext model is not a flaw of the GroundCUA dataset itself. We validate this through a controlled experiment where we sampled 10k data points from GroundCUA and three competing datasets with available bounding boxes: Aguvis, OS-Atlas, and UGround. We trained a Qwen2.5-VL-3B-Instruct baseline using the hyperparameters described in Section 4.1 and Appendix C.4, with two modifications: we adopted the simpler binary (0/1) reward formulation described in the GTA paper and extended training to 2 epochs, as we observed rewards continuing to increase after the first epoch. We report the performance across various benchmarks below:
>
> | Dataset        | SSPro | OSWorld-G | SSv2  | MMBench | UI-V |
> |----------------|-------|-----------|-------|---------|------|
> | **Baseline**   | 29.0  | 37.4      | 81.8  | 60.8    | 6.3  |
> | **Aguvis**     | 31.2  | 45.6      | 86.01 | 67.01   | 14.7 |
> | **OSAtlas**    | 30.4  | 46.4      | 62.0  | 67.7    | 14.1 |
> | **UGround**    | 33.6  | 43.8      | 89.0  | 68.8    | 16.7 |
> | **GroundCUA** | 36.8 | 48.8   | 88.9  | 70.5    | 19.2 |
> | **Imp. over baseline** | +7.8% | +11.4% | +7.1% | +9.7% | +12.9% |
>
> We observe that GroundCUA provides substantial gains of **9.8\%** on average over the Qwen2.5-VL-3B baseline. Furthermore, it achieves an average gain of 1.9% over the next best baseline (and 2.5% if we exclude the SSv2 benchmark). This validates that the GroundCUA data is highly effective for RL. We attribute this to our human-annotated labels and bounding boxes (less noise) and the rich diversity of platforms covered by our dataset.
>
> The strength of GroundCUA lies in its platform diversity (87 applications across 12 categories) and dense annotations, which offer a huge variety of UI elements for agents to learn. RL training on a very large scale (e.g., 700k samples in our case) is computationally expensive, especially in resource-constrained settings (e.g., we used 8 H100 GPUs for our experiments). Hence, we believe the diversity of GroundCUA can be effectively exploited through a careful combination of SFT, which teaches new knowledge, and RL, which helps with generalization. Future research could focus on optimizing this SFT/RL mix, which has shown promise in works like [5]. By releasing our data, we provide the necessary resources to explore this path.
>
> We have included this discussion in Appendix D.6 of the revised paper.

---

> ### Author Response · Authors · 2025-11-24
> **Response to Reviewer Xw9S (3/3)**
>
> **Q4:** *"Is there a timeline for open-sourcing the tasks and trajectories?"*
>
> **Response:** We are fully committed to open-sourcing all data, models, and code associated with this work. For immediate review, we provide anonymous links below to the raw dataset and a GitHub repository with code to visualize and inspect the data:
> - **Raw Dataset (Screenshots, Bounding Boxes, Metadata):** [Link to dataset.](https://huggingface.co/datasets/CUAResearch/GroundCUA-ICLR-anon)
> - **Visualization Repo (Sampled Data, Visualization Code, Instruction Tuning Samples):** [Link to repository.](https://anonymous.4open.science/r/GroundCUA-ICLR-anon)
>
> We are actively preparing the full dataset for public release and look forward to releasing it to the community soon.
>
>
> ## **References:**
> [1] Li et al. ScreenSpot-Pro: GUI Grounding for Professional High-Resolution Computer Use
>
> [2] Nayak et al. UI-Vision: A Desktop-centric GUI Benchmark for Visual Perception and Interaction
>
> [3] Wang et al. MMBench-GUI: Hierarchical Multi-Platform Evaluation Framework for GUI Agents
>
> [4] Yang et al. Aria-UI: Visual Grounding for GUI Instructions
>
> [5] Ma et al. Learning What Reinforcement Learning Can't: Interleaved Online Fine-Tuning for Hardest Questions

---

### Official Review · Reviewer_aAn7 · 2025-11-01

**Soundness:** 4
**Presentation:** 4
**Contribution:** 2
**Rating:** 4
**Confidence:** 4

**Summary:**

The paper introduces GROUNDCUA, a large, expert annotated desktop UI grounding dataset built from human demonstrations across 87 applications and 56K screenshots with over 3.56M element annotations. The authors convert these dense annotations into contextual instruction. Based on this dataset, a series of 3B/7B models are also trained, which are termed GROUNDNEXT. They are trained with supervised fine-tuning, followed by a light RL. On five benchmarks spanning desktop, web, and mobile, GROUNDNEXT achieves the top average SFT performance at both model sizes, with RL offering small but consistent gains.

**Strengths:**

GROUNDCUA features high quality, human verified supervision data. The elements are hand labeled from expert task demonstrations, giving reliable targets rather than noisy accessibility or synthetic signals.

The dataset contains dense and fine grained coverage of samples, with screens averaging 64 labeled elements to support precise grounding at pixel-level granularity.

The SFT model has demonstrated strong performance with modest data volume. GROUNDNEXT tops SFT baselines across five benchmarks for both 3B and 7B sizes, highlighting data quality over raw scale.

**Weaknesses:**

As a dataset paper, it would be great that the authors can provide a link to the dataset.

**Questions:**

Listed above.

---

> ### Author Response · Authors · 2025-11-24
> **Response to Reviewer aAn7**
>
> We thank the reviewer for their feedback. We agree that for a dataset paper, direct access to the data is essential for a complete review. We have provided anonymous links below to the raw dataset and a GitHub repository with code to visualize and inspect the data.
> - **Raw Dataset (Screenshots, Bounding Boxes, Metadata)**: [*Link to dataset*](https://huggingface.co/datasets/CUAResearch/GroundCUA-ICLR-anon)
> - **Visualization Repo (Sampled Data, Visualization Code, Instruction Tuning Samples)**: [*Link to repository*](https://anonymous.4open.science/r/GroundCUA-ICLR-anon)
>
> We note that the reviewer assigned a contribution score of 2 and an overall score of 4, yet the only stated weakness was the missing dataset link. If there are any other specific concerns that we have not addressed, we would be eager to hear them.
> Meanwhile, we wish to re-emphasize our main contributions, which we believe are highly significant for the community:
> 1. **A High-Quality Dataset from Expert Demonstrations (GroundCUA):** Our primary contribution is a dataset built from expert human demonstrations, rather than relying on (often incomplete) automated accessibility-tree traversal or synthetic generation. GroundCUA is the largest human-annotated desktop dataset (3.56M elements), offering significantly higher density (64 elements/screen) and resolution coverage than prior automated or synthetic datasets.
> 2. **Demonstrated Data Quality and Efficiency:** Results from our GroundNext models demonstrate that training on just 700K samples from our dataset (<10% of prior SOTA data) achieves state-of-the-art performance. Our controlled experiments in Figure 3 further reinforce this point by directly comparing SFT performance against other public datasets.
> 3. **An Open-Source Foundation for New Research:** We will release all data, models, and code. We believe the dense, high-fidelity nature of GroundCUA enables research avenues that were difficult with previous datasets, such as dense reward modeling and cross-application generalization.
>
> We hope the provided links and this summary clarify the dataset's value and its demonstrated ability to train SOTA models with 10x less data. Hence, we respectfully ask the reviewer to reconsider their evaluation.

---

### Official Review · Reviewer_AXGb · 2025-11-02

**Soundness:** 3
**Presentation:** 3
**Contribution:** 3
**Rating:** 6
**Confidence:** 3

**Summary:**

This paper tackles the grounding problem for computer-use agents. Basically helping AI systems correctly identify which UI element to click when given natural language instructions. The main contribution is GROUNDCUA, a massive dataset with 56K desktop screenshots from 87 applications. The dataset coverage is pretty broad, from office software to creative tools to development environments.

**Strengths:**

GROUNDCUA fills a major gap in desktop grounding data. Most existing datasets focus on web or mobile interfaces, but desktop apps are way more complex with tiny icons and dense layouts. The human expert approach is smart - instead of automated scraping, they had people actually use the software and annotate everything they see.

The results show that data quality beats data quantity. The two-stage approach with supervised fine-tuning plus reinforcement learning is pretty straightforward. No complex reward engineering needed.

Even though the models only see desktop data during training, they transfer surprisingly well to mobile and web interfaces. This suggests that learning good grounding skills on complex desktop environments helps with simpler interfaces too.

**Weaknesses:**

The paper describes three instruction types but doesn't analyze how they affect model performance differently. The instruction generation relies heavily on prompting Qwen2.5-VL-72B. But there's no discussion of prompt sensitivity or failure cases.

What happens when the LLM generates wrong instructions? The paper mentions using "about 100 templates" for textual elements and "120 templates" for general ones. But it doesn't say which templates work best or how template diversity impacts training.

The discrete reward function with six levels seems somewhat arbitrary. Why these specific thresholds (-0.5, -0.1, 0, 0.1, 0.5)? The paper mentions trying continuous and binary rewards but doesn't provide detailed comparisons.

The RLOO method choice over GRPO is mentioned briefly but lacks analysis. How sensitive is the approach to the group size (n=8) and batch size (64) choices? The modest RL improvements suggest the reward signal might not be optimal. More sophisticated reward formulations could yield better gains.

The error analysis in Appendix E is brief and doesn't systematically categorize failure modes across domains. The paper mentions some errors come from "limited domain knowledge", it would be helpful to explore this.

**Questions:**

Please see the weakness section above for more detailed analysis.

---

> ### Author Response · Authors · 2025-11-24
> **Response to Reviewer AXGb (1/3)**
>
> **Q1:** *"The paper describes three instruction types but doesn't analyze how they affect model performance differently. The instruction generation relies heavily on prompting Qwen2.5-VL-72B. But there's no discussion of prompt sensitivity or failure cases."*
>
> **Response:** We have run new ablation studies to precisely analyze the impact of instruction types and are happy to clarify the robustness of our generation pipeline.
>
> ## **Ablations on Instruction Type**
>
> You are correct that we have three main instruction types: Direct, Functional, and Spatial. The "Direct" category is itself broad, encompassing instructions based on descriptions, text/visuals, as well as general/heuristic templates. We provide some examples from different instructions in Appendix B and the definition of different subtypes below:
>
> **Functional Instructions:** Instructions that describe the purpose or goal of an element.
>
> **Spatial Instructions:** Instructions that locate a target element based on relative position.
>
> **Direct Instruction:**
>
> - **Description:** Dense, LLM-generated descriptions that detail an element's location,
> state, or visual attributes.
>
> - **General Templates:** Created using heuristic templates with element names inserted into
> predefined placeholders.
>
> - **Text and Visual Elements:** Instructions targeting text elements or icons.
>
> - **Miscellaneous:** Instructions that don't fall into other categories.
>
>
>
> To analyze their impact, we sampled 100k datapoints for each distinct instruction subtype and trained Qwen2.5-VL-Instruct-3B model. The results are presented below:
>
> | Data Type | SS-Pro | OS-World-G | UI-Vision | MMBench-GUI | SSv2 | Avg |
> | :--- | :--- | :--- | :--- | :--- | :--- | :--- |
> | **Functional** | 43.0 | 51.0 | 27.6 | 73.4 | 88.1 | **56.6** |
> | **Spatial** | 20.5 | 52.5  | 22.4| 67.8 | 74.1 | 47.5|
> | **Description** | 36.5 | 58.5 | 24.5 | 70.1 | 85.0 | 54.9 |
> | **General Templates** | 37.9 | 52.8 | 26.4 | 73.8 | 86.1 | 55.4 |
> | **Text and Visual** | 35.3 | 51.0 | 20.3 | 64.2 | 85.1 | 51.2 |
> | **Miscellaneous** | 32.5 | 56.3 | 24.7 | 67.1 | 85.4 | 53.2 |
>
> As the table shows, Functional instructions yield the highest average performance. We hypothesize this is because these goal-oriented instructions (e.g., "Open a new tab") closely match the tasks in the evaluation benchmarks.
> However, the other instruction types are crucial for building a robust, well-rounded agent, and a mix is necessary to prevent overfitting.
> 1. **Complementary Strengths:** While "Functional" is best on average, other types excel at specific tasks. For example, in the OSWorld-G text-recognition category, the "Text and visual" split achieves a 0.64 score, outperforming the "Functional" split's 0.60.
>
> 2. **Preventing Overfitting:** We observed that models trained on only one instruction type become brittle. For example, a model trained only on "Description" instructions sees its performance drop by 4% on ScreenSpot-Pro when the prompt "Click on the element with the following description:" is not prepended to the benchmark's instructions.
>
> Our final 700K SFT dataset is a deliberate mix to ensure the model is both high-performing and less sensitive to prompts used. Hence, when evaluating our final models, **we do not provide any prefix prompts** and evaluate directly on the instructions provided by the benchmarks.
>
> ## **Robustness of MLLM Instruction Generation Pipeline**
> Regarding the generation pipeline, we acknowledge the reviewer's concern. While our MLLM pipeline is not perfect, it is highly robust for a key reason: we are not dependent on the MLLM's open-ended knowledge. Our pipeline is highly constrained. It provides the MLLM with strong, ground-truth context, including the element's name, its bounding box, and additional screenshot context (e.g., elements around the target element for spatial instructions). The task is one of grounded rephrasing or description, not open-ended task creation.
>
> The effectiveness of this method is supported by the results in the table above: the MLLM-generated "Functional" and "Description" instructions perform better or on par with the "General Templates". These "General" templates were created using simple, hand-crafted heuristics (e.g., "Click on the following element: [element name]"). More as explained above, different types of instructions bring complementary abilities necessary for a robust agent. This indicates our MLLM-based data is high-quality and the generation process is effective. However, we agree that this could be further improved but that is beyond the scope of our paper.

---

> ### Author Response · Authors · 2025-11-24
> **Response to Reviewer AXGb (2/3)**
>
> **Q2:** *"What happens when the LLM generates wrong instructions? The paper mentions using "about 100 templates" for textual elements and "120 templates" for general ones. But it doesn't say which templates work best or how template diversity impacts training."*
>
> **Response:**
> ## **Impact of template diversity**
>
> Our goal in using diverse templates (e.g., "Click on X," "Locate X," "Find the element X") is not to find the "single best" template, but to ensure our model is a robust, general-purpose grounding agent that is not over-sensitive to a specific prompt format. This practice is common in prior work [1, 2].
>
> We do not believe one template provides any benefit over the other. Our hypothesis is that a model trained on diverse prompts will generalize better to new, unseen instructions. As shown in our previous response (Q2), a model trained only on "Description" instructions (a single type) sees its performance drop by 4% on ScreenSpot-Pro when the specific prefix prompt is removed. This demonstrates the brittleness of a model trained on a single format. Moreover, in real word agentic settings humans can make requests in different formats and we want our models to handle this effectively.
>
> ## **Wrongly generated instructions**
>
> First, we want to emphasize that the quality of our raw GroundCUA dataset is independent of the instruction-tuning data. Our primary contribution is this high-quality, human-curated raw data, which can be used with any future instruction generation pipeline.
>
> That said, our MLLM pipeline for generating the instructions is highly robust. As we noted previously (Q2), this is a constrained task, not open-ended generation. The MLLM is provided with strong, ground-truth context (the element's name, its bounding box, and surrounding context), which makes the generated instructions highly reliable. This also clearly reflects the outcome of the trained model using generated instructions across a wide variety of benchmarks and the reported result in the paper.
>
> To further verify this systematically, we performed a human evaluation on the generated instruction-tuning set. Three annotators who are not the authors of the paper annotated 100 randomly selected instructions to check for validity (i.e., whether the instruction accurately describes the element being grounded). Using a majority vote to aggregate these annotations, we found an error rate of 4%. We believe this represents a low error rate for a training dataset, highlighting the quality and reliability of our MLLM-generated instruction data.
>
> ---
>
> **Q3:** *"The discrete reward function with six levels seems somewhat arbitrary. Why these specific thresholds (-0.5, -0.1, 0, 0.1, 0.5)? The paper mentions trying continuous and binary rewards but doesn't provide detailed comparisons."*
>
> **Response:** We thank the reviewer for raising this point.
>
> The discrete reward thresholds were selected as a simple and interpretable way to partition the normalized distance range and provide meaningful gradations of quality during training. We also experimented with alternative reward schemes and found that the selected discretization performed among the strongest across benchmarks.
>
>  Furthermore, we tested reward designs ranging from 2 to 8 levels and got the best performance from the 6-level variant used in the paper. Given that this simple heuristic worked well, we did not explore more elaborate reward designs. However, we agree that exploring complex and more sophisticated reward functions on GroundCUA is an exciting future direction.
>
> We report the performance of the different reward designs in the table below and also discuss this in Appendix C.5.2 of the revised paper.
>
> | Reward                | MMBench-GUI | OSWorld-G | ScreenSpot-Pro | Avg   |
> |-----------------------|------------:|----------:|---------------:|------:|
> | (-1.0, -0.5, -0.3, -0.1, 0.1, 0.3, 0.5, 1.0)   | 80.63%      | 61.52%    | 52.31%         | 64.82%|
> | (-1.0, -0.5, -0.1, 0.1, 0.5, 1.0)      | 80.97%      | 63.12%    | 51.61%         | 65.23%|
> | (-1.0, -0.5, 0.5, 1.0)         | 80.41%      | 62.59%    | 52.56%         | 65.19%|
> | (-1.0, 1.0)            | 81.02%      | 62.06%    | 52.37%         | 65.15%|

---

> ### Author Response · Authors · 2025-11-24
> **Response to Reviewer AXGb (3/3)**
>
> **Q4:** *"The RLOO method choice over GRPO is mentioned briefly but lacks analysis. How sensitive is the approach to the group size (n=8) and batch size (64) choices? The modest RL improvements suggest the reward signal might not be optimal. More sophisticated reward formulations could yield better gains."*
>
> **Response:** We thank the reviewer for this question. Our aim in this work is to adopt a stable RL configuration rather than to optimize the RL algorithm. Following prior work such as InfiGUI-G1 [3] and GUI-G2 [4], we use a group size of n=8, which offers a good balance between stability and efficiency, and we avoid going below this value. While larger groups, such as n=32, might provide slightly stronger results, they require substantially longer training.
>
> Our ablation on GroundNext-3B (SFT) with 5K data points using the setup discussed in Section 4.1 shows that all group sizes of 4, 8, and 32 produce very similar averages, indicating that the method is not highly sensitive to the group size in this regime.
>
> | n   | ScreenSpot-Pro | ScreenSpot-v2 | OSWorld-G | UI-Vision | Avg   |
> |-----|----------------:|--------------:|----------:|----------:|-----------:|
> | 4   | 48.1          | 88.1         | 59.4    | 58.1    | 63.4  |
> | 8   | 48.6          | 87.7         | 59.0    | 57.8    | 63.3  |
> | 32  | 49.5          | 88.2         | 58.3    | 58.2    | 63.5  |
>
> We went with a batch size of 64, as higher batch sizes in combination with other parameters used to train the models sometimes led to out-of-memory errors. We agree that more advanced reward formulations or RL designs may yield larger gains, and it remains an interesting direction for future work.
>
> ---
>
> **Q5:** *"The error analysis in Appendix E is brief and doesn't systematically categorize failure modes across domains. The paper mentions some errors come from "limited domain knowledge", it would be helpful to explore this."*
>
> **Response:** We thank the reviewer for this constructive suggestion. We agree that a systematic analysis of failure modes is crucial for understanding the model's boundaries. We have expanded Appendix E in the revised paper to categorize errors into four distinct types. We summarize the key findings below (please refer to the revised manuscript for the figures):
>
> 1. **Limited Domain Knowledge (Generalization to Web/Mobile):** The reviewer correctly noted that some errors stem from "limited domain knowledge," as GroundCUA predominantly covers desktop software applications. This is most prevalent when generalizing to out-of-domain platforms. While GroundNext performs competitively on Mobile and Web benchmarks, errors often arise due to distribution shifts. Mobile interfaces, for instance, utilize vastly different aspect ratios, resolutions, and distinct UI patterns that are absent in our desktop-centric training data. Figure 10 shows some errors for the mobile platform.
>
> 2. **Localization Precision (Near-Misses):** We analyzed the magnitude of grounding errors and found that many "failures" are actually correct semantics with imperfect localization. As shown in Figure 11 of the revised paper, over 50% of the errors have a relative distance of less than 10% from the ground truth bounding box. This suggests the model successfully identifies the correct target region but occasionally lacks pixel-perfect precision. We visualize these "near-miss" examples in Figure 11 (left).
>
> 3. **Application-Specific Semantics:** We observe errors when the model encounters specialized terminology or icons in unseen software applications. For example, in our analysis of Platform VMWare in ScreenSpot-Pro (which is not present in our training data), the model struggles with niche tools that require specific software knowledge to identify (eg, "restart from CD", "snapshot details"), whereas it remains robust on generic UI elements like "Refresh", "font size".
>
> 4. **Spatial Reasoning Limitations:** Despite the inclusion of spatial data in our instruction mix (~13%), the model shows a notable performance gap when handling complex relative instructions. This is quantified in the UI-Vision benchmark, where performance drops from ~70.1% on the "Basic" category to ~49.9% on the "Spatial" category. We suspect that while the current data helps, solving this fully may require a base model with stronger inherent spatial reasoning capabilities or a higher ratio of spatial instruction tuning. In Figure 11 (right), we show an error in localising the correct cell in the Libre Office platform (from the OSW-G benchmark)
>
> ## **References:**
>
> [1] Gou et al. Navigating the Digital World as Humans Do: Universal Visual Grounding for GUI Agents
>
> [2] Xie et al. Scaling Computer-Use Grounding via User Interface Decomposition and Synthesis
>
> [3] Liu et al. InfiGUI-G1: Advancing GUI Grounding with Adaptive Exploration Policy Optimization
>
> [4] Tang et al. GUI-G2: Gaussian Reward Modeling for GUI Grounding

---

> > ### Comment · Reviewer_AXGb · 2025-11-25
> > **Response**
> >
> > Thanks for the response. I would like to maintain my ratings.

---

> > > ### Author Response · Authors · 2025-11-26
> > > **Response to Reviewer AXGb**
> > >
> > > We appreciate the reviewer for taking the time to read our detailed rebuttal. We believe **our responses have fully addressed the specific concerns raised.** We remain open to any further discussions should there be any remaining ambiguity or points of concern.

---

### Official Review · Reviewer_heBV · 2025-11-02

**Soundness:** 3
**Presentation:** 3
**Contribution:** 3
**Rating:** 6
**Confidence:** 4

**Summary:**

This paper introduces GROUND-CUA, a large-scale, human-annotated grounding dataset, and demonstrates its effectiveness using two-stage, state-of-the-art grounding models evaluated on several desktop benchmarks. The dataset offers diverse applications, detailed annotations, and broad coverage of action types and pixel densities. The paper’s introduction of the concept of phased rewards—which encourages models to gradually move toward the ground truth—is particularly innovative and interesting.

**Strengths:**

1. A new SOTA performance on grounding tasks: with only one-tenth of the training data compared to previous methods, GROUNDNEXT achieves state-of-the-art performance on desktop benchmarks while also generalizing to OOD categories, demonstrating the effectiveness and huge contribution of this grounding dataset.
2. Action and task types that match real-world computer tasks: demonstrations are collected from real-world user experiences, and the applications span diverse real-world use cases, helping to narrow the gap between synthetic and real-world grounding tasks.
3. The instructions generated for fine-tuning target diverse descriptions — including directness, intent, and location — thereby enhancing the generalization ability of the fine-tuned models.

**Weaknesses:**

1. Although the authors demonstrate their dataset's effectiveness on the grounding task, they did not show whether this grounding ability transfers to improved performance on computer-use tasks.
2. After the model underwent reinforcement learning on 10K samples, the improvement was limited, casting doubt on the authors' motivation for using the reinforcement learning step.

**Questions:**

1. Why choose RLOO rather than the normally used GRPO? Are there any preliminary experiments that show RLOO is better?

2. When does $D_{norm}$ in the reward definition exceed 0.5? Does that imply the grounding box is approximately the same size as the screenshot? Are you using the bounding box center as the reference point? Could you provide concrete examples for the different reward types?

3. I would be grateful if the authors could provide some analysis on the dataset’s scaling trend: if we sample datasets of different sizes and perform SFT on the model, will we observe a similar scaling curve in the grounding domain?

4. It would be better if the test results in the main table had confidence intervals.

---

> ### Author Response · Authors · 2025-11-24
> **Response to Reviewer heBV (1/4)**
>
> **Q1:** *"Although the authors demonstrate their dataset's effectiveness on the grounding task, they did not show whether this grounding ability transfers to improved performance on computer-use tasks."*
>
> **Response:** We evaluate GroundNext’s performance in an agentic setting. Experiments are conducted on the OSWorld-Verified [7] benchmark. We use OpenAI o3 as the planner, as our model can only ground (following the Jedi [1] setup). The planner takes task instructions and action history to generate grounding commands that our model executes to locate target UI elements on the screen. Following the setup of [1], [2], and [3], we evaluate 361 tasks (excluding Google Drive-related ones) on an Ubuntu system with a 1920×1080 resolution, running on Microsoft Azure within 10 Docker environments.
>
> | Model | OS | Office | Daily | Pro | Workflow | Overall |
> | :--- | :--- | :--- | :--- | :--- | :--- | :--- |
> | **Proprietary Models** | | | | | | |
> | OpenAI o3 (OpenAI, 2025) | 62.5 | 14.5 | 21.4 | 38.8 | 16.5 | 23.0 |
> | CUA (OpenAI, 2025) | 23.9 | 34.6 | 55.1 | 18.3 | 18.3 | 31.4 |
> | Claude-4-Sonnet (Anthropic, 2025a) | 45.8 | 39.3 | 48.1 | 59.2 | 27.9 | 41.4 |
> | Qwen3-VL-Flash (Bai et al., 2025) | 40.9 | 53.6 | 55.1 | 22.0 | 22.0 | 41.6 |
> | UI-TARS-250705 (Qin et al., 2025) | 41.7 | 50.4 | 55.7 | 51.0 | 14.7 | 41.8 |
> | Claude-4.5-Sonnet (Anthropic, 2025b) | 70.8 | 72.6 | 61.4 | 63.3 | 49.0 | 62.9 |
> | **Open-source Models** | | | | | | |
> | Qwen2.5-VL-32B (Bai et al., 2025) | 8.3 | 1.7 | 6.4 | 6.1 | 2.2 | 3.9 |
> | Qwen2.5-VL-72B (Bai et al., 2025) | 16.7 | 4.3 | 6.4 | 2.0 | 3.2 | 5.0 |
> | Kimi-VL-A3B (Kimi Team, 2025) | 12.5 | 6.0 | 21.7 | 18.4 | 1.1 | 10.3 |
> | OpenCUA-A3B (Wang et al., 2025a) | 12.5 | 16.3 | 21.7 | 46.9 | 2.2 | 17.7 |
> | UI-TARS-72B-DPO (Qin et al., 2025) | 37.5 | 19.0 | 34.6 | 63.3 | 8.3 | 27.1 |
> | OpenCUA-7B (Wang et al., 2025a) | 41.7 | 22.2 | 37.1 | 49.0 | 9.3 | 27.0 |
> | UI-TARS-1.5-7B (Qin et al., 2025) | 33.3 | 29.9 | 37.9 | 53.1 | 9.1 | 29.6 |
> | OpenCUA-72B (Wang et al., 2025a) | 58.3 | 47.0 | 53.8 | 73.5 | 20.4 | 46.1 |
> | JEDI-7B w/ o3 (Xie et al., 2025) | 50.0 | 46.1 | **61.9** | **75.5** | 35.3 | **51.0** |
> | **GroundNext-3B w/ o3 (ours)** | **62.5** | **47.0** | 55.0 | 73.5 | **36.5** | **50.6** |
>
> The results in the table above highlight GroundNext-3B's strong performance. Within its 3B parameter class, our model (50.6 Overall) significantly outperforms peers like OpenCUA-A3B (17.7) and Kimi-VL-A3B (10.3). Notably, it surpasses many larger models, including OpenCUA-72B (46.1) and proprietary APIs such as Qwen3-VL-Flash (41.6) and Claude-4-Sonnet (41.4). The comparison with JEDI-7B, which also uses the o3 planner, is particularly notable. Despite being less than half the size, our 3B model achieves a comparable overall score (50.6 vs. 51.0) and demonstrates superior performance in 3 out of 5 categories (OS, Office, and Workflow). This performance from a compact 3B model underscores GroundNext-3B's significant practical utility, presenting it as an effective and efficient solution for real-world agentic systems where inference speed and resource constraints are critical factors.
>
> We have included this analysis to Section 5.3 in the revised paper.
>
> ---
>
> **Q2:** *"After the model underwent reinforcement learning on 10K samples, the improvement was limited, casting doubt on the authors' motivation for using the reinforcement learning step."*
>
> **Response:** We thank the reviewer for this comment. We wish to clarify that our motivation for the RL step was twofold: (1) to efficiently push the performance of GroundNext models, thereby showing the effectiveness of the combination of SFT and RL with high-quality data, and (2) to empirically validate that GroundCUA can be effectively used to perform RL on grounding models. We hypothesize that the "limited" improvement is not a failure of the RL step, but rather a finding that highlights the interaction between strong SFT baselines and RL gains. We detail our analysis below.
>
> ## **High-Quality SFT Creates a "Strong Ceiling"**
>
> We hypothesize that stronger GUI models contain fewer actionable errors after the SFT stage, resulting in lower marginal benefits from subsequent RL fine-tuning—especially when the RL data is drawn from the same distribution as the SFT data. This is supported by Figure 3, where we observe that RL provides significantly larger gains for SFT models trained with other datasets compared to GroundCUA. We attribute this to the fact that GroundCUA yields a much stronger initial model; consequently, the RL stage serves as a minor refinement rather than a primary performance driver in our current setting.
>
>  ***[continued.]***

---

> ### Author Response · Authors · 2025-11-24
> **Response to Reviewer heBV (2/4)**
>
> We also see this trend in a related work, GTA1 [2]. GTA1-7B initializes its training from UI-Tars-1.5-7B, which is a powerful GUI grounding model. When we analyze the observed gains across all five major benchmarks, the average improvement is moderate, supporting the idea that the marginal return for RL on top of a strong SFT baseline is inherently challenging. We provide a detailed comparison below:
>
> | Benchmark | UI-Tars-1.5-7B | GTA-1-7B | Improvement |
> | :--- | :--- | :--- | :--- |
> | **SS-Pro** | 47.9 | 50.1 | 2.2% |
> | **OSW-G** | 64.2 | 67.7 | 3.5% |
> | **MMB-GUI** | 75.4 | 79.4 | 4.0% |
> | **SSv2** | 90.3 | 92.4 | 2.1% |
> | **UI-V** | 20.8 | 25.7 | 4.9% |
>
> We observe that the average improvement across five benchmarks is 3.3%. This demonstrates that, while RL provides a consistent lift, the gains are moderate, not universally massive. The limited marginal return observed in GroundNext is consistent with the observations for GTA-1-7B.
>
> **Important Note:** We emphasize that **we do not make a general claim that stronger GUI models cannot be effectively RL-tuned.** Our results merely provide evidence supporting the hypothesis that the marginal return is lower when a robust, high-quality SFT initialization is used, particularly for models in the 3–7B size range under the reward formulations we employ.
>
> A more detailed study spanning different architectures, various reward functions, different model sizes, and deeper RL fine-tuning is required to fully understand these interactions. This comprehensive investigation is beyond the scope of our current paper, but our results provide the initial evidence for an interesting phenomenon.
>
> ## **GroundCUA is a great source for RL fine-tuning**
>
> We clarify that the limited marginal gain observed in the final GroundNext model is not a flaw of the GroundCUA dataset itself. We validate this through a controlled experiment where we sampled 10k data points from GroundCUA and three competing datasets with available bounding boxes: Aguvis, OS-Atlas, and UGround. We trained a Qwen2.5-VL-3B-Instruct baseline using the hyperparameters described in Section 4.1 and Appendix C.4, with two modifications: we adopted the simpler binary (0/1) reward formulation described in the GTA paper and extended training to 2 epochs, as we observed rewards continuing to increase after the first epoch. We report the performance across various benchmarks below:
>
> | Dataset        | SSPro | OSWorld-G | SSv2  | MMBench | UI-V |
> |----------------|-------|-----------|-------|---------|------|
> | **Baseline**   | 29.0  | 37.4      | 81.8  | 60.8    | 6.3  |
> | **Aguvis**     | 31.2  | 45.6      | 86.01 | 67.01   | 14.7 |
> | **OSAtlas**    | 30.4  | 46.4      | 62.0  | 67.7    | 14.1 |
> | **UGround**    | 33.6  | 43.8      | 89.0  | 68.8    | 16.7 |
> | **GroundCUA** | 36.8 | 48.8   | 88.9  | 70.5    | 19.2 |
> | **Imp. over baseline** | +7.8% | +11.4% | +7.1% | +9.7% | +12.9% |
>
> We observe that GroundCUA provides substantial gains of **9.8%** on average over the Qwen2.5-VL-3B baseline. Furthermore, it achieves an average gain of 1.9% over the next best baseline (and 2.5% if we exclude the SSv2 benchmark). This validates that the GroundCUA data is highly effective for RL. We attribute this to our human-annotated labels and bounding boxes (less noise) and the rich diversity of platforms covered by our dataset.
>
> The strength of GroundCUA lies in its platform diversity (87 applications across 12 categories) and dense annotations, which offer a huge variety of UI elements for agents to learn. RL training on a very large scale (e.g., 700k samples in our case) is computationally expensive, especially in resource-constrained settings (e.g., we used 8 H100 GPUs for our experiments). Hence, we believe the diversity of GroundCUA can be effectively exploited through a careful combination of SFT, which teaches new knowledge, and RL, which helps with generalization. Future research could focus on optimizing this SFT/RL mix, which has shown promise in works like [5]. By releasing our data, we provide the necessary resources to explore this path.
>
> We have included this discussion in Appendix D.6 of the revised paper.
>
> We hope this clarifies the reviewer's reservations regarding the reinforcement learning stage in our work. We would be happy to discuss this further to address any remaining questions.

---

> ### Author Response · Authors · 2025-11-24
> **Response to Reviewer heBV (3/4)**
>
> **Q3:** *"Why choose RLOO rather than the normally used GRPO? Are there any preliminary experiments that show RLOO is better?"*
>
> **Response:** We selected RLOO based on our preliminary experiments, where it demonstrated slightly better rewards and training stability compared to GRPO. Its effectiveness has been demonstrated in related GUI grounding work, such as InfiGUI-G1 [4].
> We performed initial comparisons on an early SFT checkpoint specifically for hyperparameter tuning using 5k data samples, and it showed better validation performance. We report the performance on a few benchmarks we evaluate on:
>
> | | ScreenSpot-Pro | ScreenSpot-v2 | OSWorld-G | Avg |
> |---------|-----------------|---------------|-----------|-----|
> | **RLOO** | 49.3 | 88.3 | 59.4 | 65.7|
> | **GRPO** | 49.4 | 88.2 | 58.3 | 65.3|
>
> The results show RLOO yielded a slightly higher average performance (65.7 for RLOO vs. 65.3 for GRPO), with RLOO performing notably better on the OSWorld-G benchmark. Since the performance difference between the two algorithms is marginal, we do not claim that RLOO is categorically superior to GRPO. Still, the slight empirical advantage and consistency led us to adopt RLOO for our final experiments.
>
> We provide this explanation in Appendix C.5.3 in the revised paper.
>
> ---
>
> **Q4:** *"When does D_norm in the reward definition exceed 0.5? Does that imply the grounding box is approximately the same size as the screenshot? Are you using the bounding box center as the reference point? Could you provide concrete examples for the different reward types?"*
>
> **Response:** We thank the reviewer for this question. We want to clarify that there was a typographical error in the reward function formulation in the text, but our code implementation followed the correct logic. We have updated the manuscript with the accurate definition and highlighted all changes in red in Section 4.1 and Appendix C.4.
>
> Based on the correct reward formulation, $D_{norm}$​ will exceed 0.5 if the predicted point lies inside the bounding box and its distance to the nearest corner is greater than one quarter of the box’s diameter. Hence, $D_{norm}$​ can go above 0.5 for bounding boxes of any size. We also clarify that the bounding box edges are used as the reference points. When the predicted point is outside the box, the goal is to minimize the distance to the reference; when it is inside, the goal is to maximize the distance from the edges, encouraging the model to predict the center of the box. **Ultimately, our reward function encourages the model to predict the center of the bounding box rectangle**. As a concrete example, we have included Figure 9 in the revised paper. It shows six predicted points with varying $D_{norm}$ scores and the corresponding rewards received based on their distance from the ground truth bounding box.
>
> We briefly restate the correct reward formulation for clarity. When the predicted point lies inside the ground-truth bounding box, the denominator $D_{ref}$ becomes half of the box’s diameter ($0.5⋅diam(B)$), which corresponds to the distance from a corner to the center. The numerator is the shortest distance from the predicted point to any corner, keeping the normalized distance in the range $[0,1]$. When the point lies outside the bounding box, the numerator $D(\hat{p},B)$ is the shortest distance from the predicted point to the box edges, and the denominator is the largest possible distance a point outside the box can have relative to it ($D_{max}​(B,I)$). The formulation originally written in the paper corresponds to the outside-box case, where $D_{norm}$​ ranges from $[−1,0]$. The corrected description now matches the implementation and ensures a stable, scale-invariant reward across both regimes.

---

> ### Author Response · Authors · 2025-11-24
> **Response to Reviewer heBV (4/4)**
>
> **Q5:** *"I would be grateful if the authors could provide some analysis on the dataset’s scaling trend: if we sample datasets of different sizes and perform SFT on the model, will we observe a similar scaling curve in the grounding domain?"*
>
> **Response:** To analyze the scaling behavior in the grounding domain, we conducted an ablation study using Qwen2.5-VL-Instruct 3B model (as the time permitted us during this rebuttal period). We sampled subsets of 100k (smaller region) and 350k (mid region) instructions from our full 700k SFT dataset and trained models using identical hyperparameters.
>
> | Size | SS-Pro | OS-World-G | MMBench-GUI | UI-Vision | SS-v2 | Average |
> | :--- | :---: | :---: | :---: | :---: | :---: | :---: |
> | **100k** | 41.7 | 54.6 | 72.5 | 29.8 | 87.4 | 57.2 |
> | **350k** | 44.5 | 57.3 | 72.9 | 39.5 | 87.0 | 60.2 |
> | **700k** | 48.6 | 62.2 | 75.5 | 58.2 | 87.3 | 66.4 |
>
> We observe consistent performance gains across most benchmarks as the data size increases. This is generally expected for a stable training setup — adding more data improves performance to a margin. Further, we attribute this continued improvement to the rigorous deduplication process applied during dataset creation. Because we actively filter highly redundant elements, adding more data points introduces new, non-redundant visual and semantic information rather than repetitive noise. The scaling is most pronounced on UI-Vision, where performance nearly doubles from 100k to 700k (29.8 to 58.2). As noted in the paper, UI-Vision is closest to our training distribution, allowing the model to fully leverage the additional data to master these specific layouts and element types.
>
> We include the scaling discussion in Appendix D.7 in the revised paper.
>
> ---
>
> **Q6:** *"It would be better if the test results in the main table had confidence intervals."*
>
> **Response:** We understand the importance of reporting statistical variance. However, we have chosen to report single-run scores for the following reasons:
>
> 1. **Deterministic Inference:** We utilize greedy decoding with temperature=0 for all our evaluations. Since we are evaluating on fixed, offline benchmarks, our inference process is mostly deterministic and reproducible.
> 2. **Compute Constraints:** Reporting confidence intervals for model performance would require training multiple models with different random seeds. As detailed in our paper, our pipeline involves both SFT and RL stages. Retraining the 3B and 7B models multiple times (e.g., at least 3 runs each) to generate error bars for all tables is computationally infeasible during the rebuttal period, given our resource constraints and the experiments we had to run to address other rebuttal questions.
> 3. **Alignment with Related Work:** We followed the reporting methodology standard in recent related works on GUI agents (e.g., [1], [2], [3], [6]), which typically report single-run performance, most likely due to similar constraints.
>
> If we have misunderstood the reviewer's request, please let us know.
>
> We hope that our clarifications have resolved any reservations the reviewer had about our paper. We respectfully ask the reviewer to reconsider their evaluation.
>
> ## **References:**
>
> [1] Xie et al. Scaling Computer-Use Grounding via User Interface Decomposition and Synthesis
>
> [2] Yang et al. GTA1: GUI Test-time Scaling Agent
>
> [3] Wang et al. OpenCUA: Open Foundations for Computer-Use Agents
>
> [4] Liu et al. InfiGUI-G1: Advancing GUI Grounding with Adaptive Exploration Policy Optimization
>
> [5] Ma et al. Learning What Reinforcement Learning Can't: Interleaved Online Fine-Tuning for Hardest Questions
>
> [6] Qin et al. UI-TARS: Pioneering Automated GUI Interaction with Native Agents
>
> [7] Xie et al. OSWorld: Benchmarking Multimodal Agents for Open-Ended Tasks in Real Computer Environments

---

> > ### Comment · Reviewer_heBV · 2025-11-25
> >
> > I really appreciate the authors' reply; it has largely resolved my questions. I now understand the effectiveness of GroundCUA on CUA tasks, have a clearer sense of the reward format, and see the advantages of RLOO. The clarification about the marginal return of RL after SFT is convincing — I now believe the limited improvement is likely due to similar data distributions and a strong SFT starting point, rather than the data quality itself. I will increase my score accordingly.

---

### Author Response · Authors · 2025-11-24
**General Response to all Reviewers**

We thank all reviewers for their thoughtful and constructive feedback. We are very grateful that the reviewers recognized **GroundCUA** for **filling a critical gap** in desktop grounding resources with **high-quality, dense, and expert-verified annotations** that capture **real-world task complexity** [Reviewers AXGb, aAn7, Xw9S]. They further highlighted the dataset's **exceptional sample efficiency**, noting that it enables GroundNext to achieve **state-of-the-art performance** and **robust cross-domain generalization** using only a fraction of the data required by previous methods [Reviewers heBV, AXGb, aAn7, Xw9S].

We are committed to addressing the questions posed by the reviewers and have updated the manuscript based on the questions so far. We summarize the changes we have made to the paper here:

1. Provided anonymous links to the GroundCUA dataset *(aAn7, Xw9S)* and the corresponding GitHub repository for visualization. (Link to [dataset](https://huggingface.co/datasets/CUAResearch/GroundCUA-ICLR-anon) and [repository](https://anonymous.4open.science/r/GroundCUA-ICLR-anon).)
2. Included new agentic results *(heBV)* for the **OSWorld-Verified** benchmark in Section 5.3, demonstrating performance transfer from benchmarks to real-world tasks.
3. Provided a detailed analysis of instruction tuning data *(AXGb)*, including ablation studies on instruction types (Appendix B.5), robustness analysis of the MLLM pipeline (Appendix B.6), and data quality assessment via a human study (Appendix B.6).
4. Included an in-depth analysis of RL implementation details (Appendix C.4) and the rationale for choosing various hyperparameters (e.g., RLOO vs. GRPO) in Appendix C.5 *(heBV, AXGb, Xw9S)*.
5. Updated the discussion on **RL gains**, providing extensive analysis in Appendix D.6 to address the "limited improvement" observation for our RL experiments *(heBV, Xw9S)*.
6. Provided dataset **SFT scaling trends** *(heBV)* for the 3B models using GroundCUA subsets, confirming the positive scaling behavior in Appendix D.7.
7. Expanded the discussion on **GroundNext error analysis** (Appendix E), systematically categorizing failure modes *(AXGb)*.

We believe that this discussion has significantly strengthened the manuscript. We are happy to discuss and clarify any remaining questions the reviewers have.

---

### Author Response · Authors · 2025-12-04
**Summary of Discussions**

We appreciate the AC’s support and effort in navigating the current review situation. To assist in the final decision-making, we provide a brief recap of the rebuttal below:

**General Overview:** We have included a comprehensive summary of our detailed rebuttal responses in the General Response to all Reviewers below.

**Response to Reviewer heBV:** We provided extensive justification for the RL gains (a concern also shared by Reviewer Xw9S) and addressed all other questions. Following this clarification, Reviewer heBV **raised their score.**

**Addressing Other Concerns:** We provided detailed experimental justifications and design ablations to address comments from Reviewers Xw9S and AXGb (who acknowledged our rebuttal and raised no further questions), and addressed Reviewer aAn7’s **only concern** by providing the dataset URL. We understand that the recent interruption likely prevented other reviewers from updating their scores, but we believe our responses fully resolved their stated concerns.

We want to emphasize that the reviewers consistently praised GroundCUA for its high quality and sample efficiency. We believe the additional experiments have significantly strengthened the paper, and we look forward to presenting the improved version of this work.

Regards,

Authors

---

### Meta-Review · Area_Chair_Jti5 · 2026-01-06

**Summary:**

This paper addresses the GUI grounding problem in computer‑use agents (i.e., mapping natural‑language instructions to on‑screen elements) by introducing the GroundCUA dataset and the GroundNext model series. GroundCUA is currently the largest, expert‑human‑demonstrated desktop GUI grounding dataset, containing 56K screenshots from 87 applications and 3.56M human‑verified annotations. Using this dataset, the authors train GroundNext models with 3B and 7B parameters. Through two‑stage training (supervised fine‑tuning followed by reinforcement learning), the models achieve state‑of‑the‑art performance on five benchmarks while using only about one‑tenth of the training data of prior work, and demonstrate strong cross‑domain generalization.

**Reviewer Concerns:**

- **Reviewer heBV:** [Initial score 6 → inclined to increase] Initially had concerns about agentic transferability and RL gains, but after seeing the additional experiments and explanations, explicitly stated that the issues were resolved and would raise the score.
- **Reviewer Xw9S:** [Score 6] Acknowledged the value of the dataset and its sample efficiency, raised questions about why full‑trajectory data were not used and the omission of related work (Aria‑UI), which the authors addressed.
- **Reviewer AXGb:** [Score 6] Recognized that the work fills a gap in desktop data, but had questions about the impact of instruction types and RL hyper‑parameters (RLOO vs. GRPO). After the rebuttal, indicated they would maintain their original score.
- **Reviewer aAn7:** [Score 4] Considered the data quality excellent, but gave a low overall score due to the missing dataset link. The authors have since provided the link.
- **Main Points of Disagreement:** No major disagreements. Discussions centered primarily on the magnitude of RL gains and methodological details, most of which were clarified in the rebuttal.

**Reviewer Scores:**

This paper makes a substantial resource contribution (GroundCUA) to the GUI grounding field and presents a strong baseline model. Reviewers unanimously acknowledge that the dataset fills a critical gap in desktop environment data and that the data quality is very high. Regarding scores:
1. Reviewer aAn7’s score of 4 was primarily based on a single weakness (“missing dataset link”), which has now been resolved.
2. Reviewer heBV explicitly stated they would raise their score after the rebuttal.
3. Reviewer AXGb maintained a score of 6 (above the borderline‑accept threshold) and raised no grounds for rejection.

---

### Decision · Program_Chairs · 2026-01-26

Accept (Poster)